# Structure-Augmented Reasoning Generation

## Abstract

Recent advances in Large Language Models (LLMs) have significantly improved complex reasoning capabilities. Retrieval-Augmented Generation (RAG) has further extended these capabilities by grounding generation in dynamically retrieved evidence, enabling access to information beyond the model's training parameters. However, while RAG addresses knowledge availability, standard pipelines treat retrieved documents as independent, unstructured text chunks, forcing models to implicitly connect information across fragmented context. This limitation becomes critical for multi-hop queries, where answering correctly requires synthesizing information scattered across different documents. We present Structure-Augmented Reasoning Generation (SARG), a post-retrieval framework that addresses this gap by materializing explicit reasoning structures from retrieved context. SARG operates in three stages: extracting relational triples from retrieved documents via few-shot prompting, organizing these triples into a domain-adaptive knowledge graph, and performing multi-hop traversal to identify relevant reasoning chains. These chains, along with their associated text chunks, are then integrated into the generation prompt to explicitly guide the model's reasoning process. Importantly, SARG doesn't require custom retrievers or domain-specific fine-tuning. Instead, it functions as a modular layer compatible with all existing RAG pipelines. Extensive experiments on open-domain QA benchmarks and specialized reasoning datasets in finance and medicine demonstrate that SARG significantly outperforms state-of-the-art flat-context RAG baselines in both factual accuracy and reasoning coherence. Furthermore, by surfacing the exact traversal paths used during generation, SARG provides fully traceable and interpretable inference.

## 1 Introduction

Large Language Models (LLMs) have demonstrated remarkable capabilities across natural language tasks such as question answering Kamalloo et al. (2023); Lazaridou et al. (2022), summarization Kotkar et al. (2024); Zhang et al. (2024), and information extraction Aggarwal et al. (2025); Fornasiere et al. (2024). However, because knowledge is encoded in model parameters, LLMs face inherent limitations: they cannot access information beyond their training data, struggle in specialized domains requiring technical expertise, and often produce hallucinations, plausible sounding but factually incorrect outputs Wang et al. (2023a); McKenna et al. (2023). Retrieval-Augmented Generation (RAG) has emerged as a key tool to address these limitations, grounding LLM outputs in dynamically retrieved, external evidence Lewis et al. (2020); Borgeaud et al. (2022); Guu et al. (2020).

While RAG addresses the problem of knowledge availability, it fails to address the problem of *knowledge synthesis* Gao et al. (2025). Standard RAG pipelines treat retrieved passages as independent, unstructured text blocks, forcing models to implicitly connect information scattered across fragmented context Ko et al. (2025); Brådland et al. (2025). This limitation becomes critical for multi-hop queries, where answering correctly requires reasoning over multiple documents to bridge disparate facts. In such settings, models succumb to the "lost-in-the-middle" phenomenon, failing to capture cross-document dependencies and yielding shallow synthesis, even when all necessary information is present in the retrieved context Liu et al. (2024); Jin et al. (2024).

Recent efforts have attempted to address this synthesis gap by adding structure to the retrieval process. Methods like GraphRAG Edge et al. (2024) construct global knowledge graphs by indexing entire document corpora upfront, extracting entities and relationships across all texts before any queries can be answered. However, corpus-wide indexing comes with significant practical drawbacks: the pre-processing phase often requires hours or days for large collections, and any changes to the corpus necessitate complete re-indexing Chen et al. (2025). This upfront investment effectively replaces traditional retrieval mechanisms with graph-based lookup, reducing the flexibility that makes RAG systems attractive for dynamic knowledge sources. An alternative is to leverage generic broad-coverage knowledge graphs like Wikidata Vrandečić & Krötzsch (2014) or ConceptNet Speer et al. (2017), which provide pre-built structure without custom indexing. However, these approaches lack the granular, context-specific relationships that specialized queries demand.

To address these limitations, we introduce Structure-Augmented Reasoning Generation (SARG), a lightweight, post-retrieval framework that constructs explicit reasoning structures on-demand from retrieved documents. Unlike methods that require corpus-wide indexing, SARG operates exclusively on whatever passages a retrieval system returns, whether from sparse retrievers, dense retrievers, or curated domain-specific collections. This makes SARG compatible with any existing RAG pipeline as a drop-in reasoning layer. Our framework works in three stages. First, SARG leverages few-shot LLM capabilities to extract relational triples from the retrieved passages, capturing key implicit and explicit connections without domain-specific training. Second, these triples are organized into a query-specific knowledge graph, with entity resolution linking references to the same concepts across documents. Third, at query time, bidirectional graph traversal identifies multi-hop reasoning chains connecting query entities to potential answers. These chains, along with their source text, are serialized and integrated into the generation prompt, providing the LLM with explicit structural guidance. By constructing structure only from retrieved content rather than indexing entire corpora, SARG maintains the flexibility of traditional RAG while injecting the explicit reasoning capabilities of graph-based methods.

We evaluate SARG on established multi-hop reasoning benchmarks, HotPotQA Yang et al. (2018) and MuSiQue Trivedi et al. (2022), as well as two custom domain-specific corpora we constructed to test long-form reasoning: Bitcoin Price Fluctuations (BP) and Gaucher Disease (GD). Unlike fact-specific multi-hop datasets, these custom benchmarks require synthesizing complex explanatory answers from specialized technical literature. Extensive experiments demonstrate that SARG consistently outperforms flat-context RAG baselines in both factual accuracy and reasoning coherence. Furthermore, by surfacing the explicit traversal paths used during generation, SARG provides fully traceable and interpretable inference.

## 2 Related Work

### 2.1 Reasoning Limitations in Standard RAG

While Retrieval-Augmented Generation (RAG) effectively grounds LLMs in external documents, standard pipelines rely on implicit synthesis across flat context windows. Recent work identifies severe limitations in this paradigm: information fragmentation Jiang et al. (2025a) and the "lost-in-the-middle" phenomenon Liu et al. (2024) cause models to miss cross-document dependencies even when all necessary information is present. Attempts to address this via memory-based compression Qian et al. (2024) or long-context fine-tuning Wang et al. (2024) treat symptoms rather than the underlying issue: implicit reasoning over unstructured text is cognitively demanding for current architectures. SARG directly addresses this by materializing explicit reasoning structures from retrieved documents.

### 2.2 Corpus-Wide Indexing Methods

A recent line of work constructs knowledge graphs over entire corpora before queries can be answered. GraphRAG Edge et al. (2024) indexes complete corpora into hierarchical community structures with pre-generated summaries for global question answering. LightRAG Guo et al. (2024) builds dual-level (entity and chunk) graphs requiring full corpus traversal during indexing. HippoRAG Jimenez Gutierrez et al. (2024) constructs open knowledge graphs from entire corpora, applying Personalized PageRank for associative retrieval while maintaining global graph state. HyperGraphRAG Luo et al. (2025) extends this paradigm

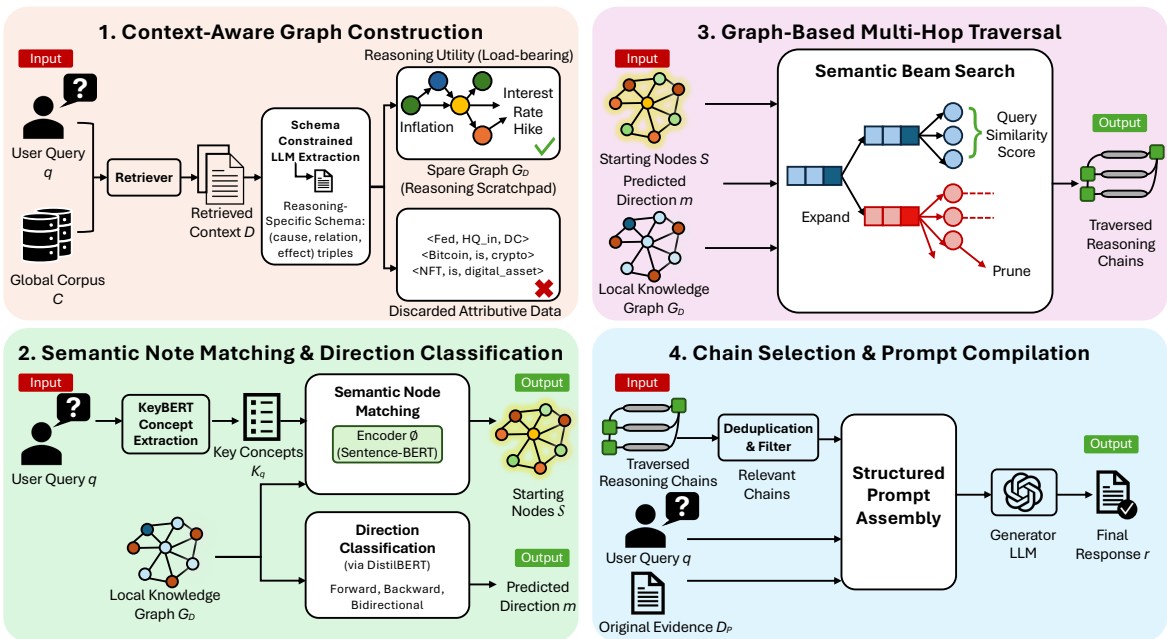

Figure 1: Overview of SARG. **Step 1:** Retrieved documents are processed to extract reasoning-relevant schema components ⟨cause, relation, effect⟩ and their context, forming a sparse graph where nodes represent concepts and edges capture their relationships. **Step 2:** Key concepts from the user query are extracted and matched to starting nodes in the local knowledge graph; our direction classifier predicts whether to traverse forward, backward, or bidirectionally based on the query structure. **Step 3:** Starting from predicted nodes, semantic beam search expands through the graph by evaluating query similarity scores at each hop, pruning low-relevance paths while retaining high-confidence reasoning chains. **Step 4:** Traversed reasoning chains are deduplicated and filtered for relevance, then combined with original evidence to assemble a structured prompt that provides attribution-aware, distilled context to the generator LLM for final response generation.

by incorporating hypergraph structures to model higher-order relationships among entities and documents, enabling richer multi-hop reasoning across large corpora.

While demonstrating strong performance, these methods face fundamental limitations. Expensive upfront processing requires $O(|C|)$ complexity where $C$ is the entire corpus, often taking hours or days for large collections. Dynamic corpus challenges arise because adding new documents requires complete re-indexing of the graph structure. The architectural inflexibility here stems from replacing traditional retrieval with graph lookup rather than augmenting it. In addition, these approaches remain fundamentally associative. By merging semantic proximity with logical implication, they fail to distinguish between relatedness and the specific causality between entities that is required for rigorous reasoning capabilities.

## 2.3 Query-Time Structuring Methods

An alternative category structures information dynamically during inference. StructRAG Li et al. (2024) employs a hybrid router that classifies queries and converts retrieved text into appropriate formats (tables, knowledge graphs, code) based on query type. Structure-R1 Wu et al. (2025) uses reinforcement learning to learn optimal structuring policies for different reasoning patterns.

While avoiding corpus-wide indexing, these methods face distinct challenges. StructRAG requires a trained classifier for format selection and focuses on representation choice rather than reasoning chain extraction. Structure-R1 needs domain-specific RL training, limiting generalization to new domains.

## 2.4 Contributions

SARG occupies a distinct point in the RAG space by combining the interpretability and depth of graph-based reasoning with the flexibility and modularity of retrieval pipelines. Our primary contributions are:

**Retriever-Agnostic & Post-Retrieval Reasoning Layer.** In contrast to corpus-wide graph construction methods that *replace* retrieval with rigid global indexing, SARG introduces a lightweight, inference-time structuring module that operates strictly downstream. By inducing structure solely over the documents returned by a classical or advanced retrieval method, such as sparse retrieval (e.g., BM25 Robertson et al. (2009)), dense retrieval (e.g., DPR Karpukhin et al. (2020), Contriever Izacard et al. (2021)), or reinforcement learning methods (e.g., s3 Jiang et al. (2025b)), SARG augments existing pipelines without requiring modification to underlying indices. This design ensures immediate compatibility with dynamic corpora.

**Focused Extraction of Reasoning-Specific Triples.** We show that instruction-tuned LLMs can reliably extract high-precision causal and logical triples from technical text without domain-specific fine-tuning or reinforcement learning. Unlike OpenIE methods that rely on syntactic parsing, often yielding noisy, surface-level relations, SARG enforces a reasoning-centric schema. By strictly targeting causal, procedural, and dependency relations, our approach acts as a semantic filter, enabling SARG to generalize across diverse domains (e.g., finance, medicine) while avoiding the combinatorial explosion typical of OpenIE-style Fader et al. (2011); Angeli et al. (2015) extraction.

**Bidirectional Multi-Hop Chain Discovery.** Given a query, SARG performs targeted traversal over the constructed reasoning graph, exploring both forward (cause → effect) and backward (effect → cause) directions. This bidirectional traversal recovers implicit intermediate steps that are not mentioned in the query but are necessary to connect retrieved evidence into coherent multi-hop explanations.

**Transparent and Verifiable Reasoning via Serialized Chains.** SARG produces answers accompanied by the exact reasoning chains used during inference, serialized directly from graph traversal. This yields fully interpretable outputs, enabling verification, debugging, and trust – capabilities that are unavailable in flat-context RAG or global graph systems that rely on summarization or attention patterns.

# 3 Methodology

## 3.1 Problem Formulation

In the standard Retrieval-Augmented Generation (RAG) paradigm, a retriever first identifies a relevant subset of documents $\mathcal{D}$ from a global corpus $\mathcal{C}$ based on a user query $q$. Conventional approaches generate a response $r$ by conditioning directly on this flat, unstructured text: $r = \text{LLM}(q, \mathcal{D})$.

SARG introduces an intermediate structured reasoning layer to bridge information gaps within $\mathcal{D}$. We formulate the SARG inference pipeline as a four-step process: (1) context-aware graph construction to induce a local graph $G_{\mathcal{D}}$ from $\mathcal{D}$; (2) semantic node matching and direction classification to identify entry points and traversal goals; (3) graph-based multi-hop traversal to extract candidate reasoning paths; and (4) chain selection and prompt compilation. The final response is generated by conditioning on the output of these steps: $r = \text{LLM}(q, \mathcal{P}, \mathcal{D}_{\mathcal{P}})$. Here, $\mathcal{P}$ represents the structured paths extracted from the graph (the reasoning), and $\mathcal{D}_{\mathcal{P}} \subseteq \mathcal{D}$ represents the specific source text segments corresponding to those paths.

## 3.2 Context-Aware Graph Construction

To operationalize this local construction, we employ a specialized extraction process. Unlike generic Open Information Extraction (OpenIE) Fader et al. (2011); Angeli et al. (2015) which retains all semantic relationships, we enforce a strict filtering mechanism designed purely for multi-hop reasoning.

### 3.2.1 Reasoning-Specific Schema

Standard knowledge graphs (e.g., Wikidata Vrandečić & Krötzsch (2014)) and recent graph-augmented retrieval architectures utilize a static schema, capturing attribute data such as $\langle \text{Obama}, \text{born\_in}, \text{Hawaii} \rangle$. While useful for fact retrieval, these static links don't support the dynamic chaining needed for complex QA.

In contrast, SARG enforces a reasoning-specific schema. We redefine the graph topology to capture logical flow rather than static properties by forcing extraction of $\langle \text{cause}, \text{relation}, \text{effect} \rangle$ triples that represent conditional dependencies, procedural sequences, and causal chains conducive to multi-hop reasoning. Nodes ($V$) represent logical states, conditions, or concepts (e.g., "inflationary pressure," "interest rate hike"), rather than just named entities. Edges ($E$) represent logical dependencies, mechanisms, or temporal sequences (e.g., "requires," "precedes").

For each document $d_i \in \mathcal{D}$, we extract triples $T_i$ constrained to our schema:

$$T_i = \text{LLM}_{\text{extract}}(d_i, P_{\text{reasoning\_schema}}) \tag{1}$$

where $P_{\text{reasoning\_schema}}$ denotes the prompt instruction set that encodes the topological constraints for logical states and dependencies defined above. This schema ensures that every edge in $G_{\mathcal{D}}$ represents a valid step in an inference chain, transforming the graph from a database of facts into a map of logical chains.

### 3.2.2 Graph Sparsity

A critical advantage of our schema is graph sparsity. General Open Information Extraction (OpenIE) typically results in dense, noisy graphs where entities are connected by trivial relations (e.g., "is a", "has"), leading to combinatorial explosion during traversal.

Because SARG filters strictly for reasoning utility, the resulting graph is intentionally sparse. We identify that for multi-hop reasoning tasks, sparsity is a key feature. It acts as a natural filter, retaining only the "load-bearing" structures (e.g., "causes", "requires", "followed by"), specifically the causal, conditional, and sequential dependencies required to bridge premises to conclusions, while discarding attributive data. This results in a graph where the number of edges scales linearly with logical steps in the text, maintaining reasoning depth without exploding graph size.

### 3.2.3 Graph Aggregation and Weighting

The complete set of extracted triples $T = \bigcup_i T_i$ forms a directed graph $G_{\mathcal{D}} = (V, E)$. We define the vertex set $V$ and edge set $E$ as:

$$V = \{v \mid \exists \langle v, r, v' \rangle \in T \lor \langle v', r, v \rangle \in T\}, \quad E = \{(u, v) \mid \langle u, r, v \rangle \in T\} \tag{2}$$

This construction produces a focused "reasoning scratchpad", a temporary, context-bound structure optimized solely for the current reasoning process.

### 3.3 Semantic Node Matching

To bridge the gap between the user's query $q$ and the graph $G_{\mathcal{D}}$, we identify entry nodes by anchoring traversal to specific query concepts. Rather than embedding the full query string, we first distill it into atomic concepts using KeyBERT Grootendorst (2020). This yields a set of key phrases $K_q = \{k_1, \ldots, k_m\}$.

We encode each extracted concept $k \in K_q$ and all graph nodes $v \in V$ into a shared embedding space using the encoder $\phi$ (Sentence-BERT Reimers & Gurevych (2019)), such that $e_k = \phi(k)$ and $e_v = \phi(v)$.

The set of starting nodes $S$ is then determined by finding semantic alignment with any of the extracted query concepts:

$$S = \left\{ v \in V \mid \max_{k \in K_q} \left( \frac{e_k \cdot e_v}{\|e_k\| \|e_v\|} \right) > \tau \right\} \tag{3}$$

This keyword-focused matching ensures that traversal initiates from nodes directly relevant to the core entities in $q$, preventing the search from being distracted by irrelevant query tokens or framing text.

### 3.4 Direction Classification via Knowledge Distillation

A critical component for efficient traversal is determining the logical flow of reasoning required by the query. Blind bidirectional traversal can introduce irrelevant noise and exponential complexity Barker & Korf (2015); Chen et al. (2017). To address this, we categorize the intent $m$ into a set of discrete reasoning directions $\mathcal{K} = \{\text{Forward, Backward, Bidirectional}\}$. While recent work demonstrates that LLMs excel at causal classification Wang et al. (2023b); Kiciman et al. (2023), relying on them at inference time introduces significant latency bottlenecks.

To eliminate this overhead without sacrificing accuracy, we employ a knowledge distillation approach. We train a lightweight student model, `distilbert-base-uncased` Sanh et al. (2019), to replicate the decision boundary of a teacher LLM (`GPT-4o-mini`). We constructed a synthetic training set of 1,448 causal reasoning queries generated by the teacher model, balanced across the three direction classes.

Formally, given a query $q$, the optimal direction $m$ is predicted by the student model $f_\theta$ by maximizing the conditional probability over the class set $\mathcal{K}$:

$$m = \underset{k \in \mathcal{K}}{\operatorname{argmax}} P_\theta(k \mid q) \tag{4}$$

where $P_\theta(k \mid q)$ denotes the probability score assigned to class $k$ by the model parameters $\theta$.

This architectural choice transforms direction classification from a system bottleneck into a negligible component. Our distilled model achieves 99.13% test accuracy, matching the teacher model's fidelity. Crucially, it reduces inference latency from ~1,485ms (`GPT-4o-mini`) to ~3ms (`DistilBERT`), a 495× speedup. This optimization allows SARG to prune the search space and ensure that retrieved chains align with the logical directionality of the user's question without incurring the cost of an additional LLM call. Comprehensive implementation details, including the synthetic data generation prompts, training hyperparameters, and the "Sim-to-Real" generalization study on real-world queries, are provided in Appendix B.

### 3.5 Graph-Based Multi-Hop Traversal

Using the identified starting nodes $S$ and the predicted direction $m$, we perform a Semantic Beam Search Dong & Lapata (2016); Herzig & Berant (2017) to identify high-utility reasoning chains. Unlike exhaustive Depth-First Search Hopcroft & Tarjan (1971), which scales exponentially and blindly explores irrelevant subgraphs, our beam search strategy leverages the query encoding to prioritize paths that remain semantically grounded to the user's intent.

We define a reasoning chain $c$ of length $L$ as a sequence of nodes $[v_1, v_2, \ldots, v_L]$. The traversal operates iteratively. At each step $t$, we expand the frontier of partial chains by following edges consistent with the predicted direction $m$ (successors for forward, predecessors for backward).

To guide the beam, we employ a scoring function based on the semantic similarity between the query embedding $e_q$ and the node embeddings $e_v$, computed via `all-MiniLM-L6-v2` Reimers & Gurevych (2019). To favor chains where every step is relevant, rather than chains with a single high-scoring outlier, we calculate the chain score $S_{chain}$ as the running average of node similarities along the path.

When extending a chain $c_{t-1}$ with score $S_{t-1}$ to a new node $v_t$, the new cumulative score is computed as:

$$S_t = \frac{S_{t-1} \cdot (t-1) + \operatorname{sim}(e_q, e_{v_t})}{t} \tag{5}$$

where $\operatorname{sim}(a, b) = \frac{a \cdot b}{\|a\| \|b\|}$ denotes cosine similarity. Note that the starting node $v_1$ serves as the anchor and is not included in the scoring average, focusing the metric on the traversed path.

At each depth, we prune the search space by retaining only the top-$\beta$ candidate chains based on $S_t$, where $\beta$ is the beam width. This reduces the traversal complexity from $O(b^d)$ to $O(\beta \cdot d)$, where $b$ is the average branching factor and $d$ is the depth.

### 3.6 Chain Selection and Prompt Compilation

The beam search terminates when chains reach a dead end. Post-traversal, we apply a deduplication filter to remove exact duplicates and prune chains that exist as sub-paths of longer, higher-scoring chains (e.g., if $A \to B$ and $A \to B \to C$ are both found, we retain the latter).

The surviving chains $\mathcal{C}_{relevant}$ are serialized into natural language and integrated into the final generation prompt $P_{\text{gen}}$ alongside the system instruction $P_{\text{inst}}$ and the associated unstructured evidence $\mathcal{D}_{\mathcal{P}}$ (the raw text segments from which the chains were derived):

$$P_{\text{gen}} = P_{\text{inst}} \oplus P_{\text{query}} \oplus \text{Serialize}(\mathcal{C}_{relevant}) \oplus \mathcal{D}_{\mathcal{P}} \tag{6}$$

This unified prompt structure forces the generator to explicitly attend to the traversed reasoning paths, effectively providing a "scaffold" for the final answer.

## 4 Experiments

**Research Questions.** We evaluate SARG on both specialized datasets and standard multi-hop reasoning benchmarks to address the following research questions:

- RQ1: How does SARG perform compared to state-of-the-art methods across diverse reasoning tasks?
- RQ2: How does SARG's automated KG construction compare to expert annotated KGs?
- RQ3: Which components of SARG contribute most significantly to its effectiveness?

### 4.1 Experimental Settings

#### 4.1.1 Datasets

Our experimental design reflects realistic RAG deployment scenarios where systems typically retrieve 10-50 documents. We evaluate on four datasets spanning domain-specific reasoning and general multi-hop QA:

**Custom Domain-Specific Datasets.** To evaluate SARG on specialized reasoning tasks requiring synthesis of technical information, we constructed two focused datasets simulating post-retrieval settings with 20 high-quality documents each. The Bitcoin Price Fluctuations (BP) dataset targets financial reasoning, capturing narratives around price swings and shifts in market sentiment. We constructed the dataset to cover events from late 2024, well after the training cutoff of most frontier LLMs, to test the effects of structured reasoning over retrieved documents without influences from parametric knowledge. The Gaucher Disease (GD) dataset covers biomedical reasoning, focusing on mechanisms underlying Gaucher disease pathology. Although GD is a well-studied condition Pastores & Hughes (2018), its rarity means LLMs typically have limited exposure to it during training, making retrieval-grounded reasoning essential.

**Multi-Hop QA Benchmarks.** To assess performance on standard reasoning benchmarks, we evaluate on two widely-adopted datasets: HotPotQA Yang et al. (2018), and MuSiQue Trivedi et al. (2022).

#### 4.1.2 Compared Methods

We benchmark SARG against a comprehensive set of baselines spanning direct generation, retrieval-augmented approaches, and graph-RAG methods:

**Standard Baselines:**

- Direct: Direct prompting with the question only, without retrieved context.
- Standard RAG Lewis et al. (2020): Retrieval-augmented pipeline that chunks documents, embeds them using a dense retriever, and concatenates top-$k$ relevant chunks as context.
- CoT-RAG Wei et al. (2022): Combines RAG with chain-of-thought prompting, instructing the LLM to reason step-by-step over retrieved context.

**Graph-Enhanced RAG Baselines:**

- GraphRAG Edge et al. (2024): Organizes documents into knowledge graphs via entity extraction, retrieves connected subgraphs, and uses community detection for context construction.
- HippoRAG2 Gutiérrez et al. (2025): Leverages hypergraph structures and Personalized PageRank for multi-hop evidence aggregation with memory-like graph indexing.
- HyperGraphRAG Luo et al. (2025): Extends graph-based RAG to n-ary relational facts via hypergraph representations beyond binary triples.
- StructRAG Li et al. (2024): Prompts LLMs to select a structure type from a predefined set (graphs, tables, catalogs, algorithms), then transforms documents into the chosen structure.

To ensure fair comparison with prior work, we evaluate methods using their originally reported model configurations. Standard baselines and SARG are evaluated across three open-source models: `Qwen2.5-7B-Instruct` Team Qwen (2024), `LLaMA-3.1-8B-Instruct` Grattafiori et al. (2024), and `DeepSeek-R1-Distill-LLaMA-8B` Guo et al. (2025). Graph-enhanced RAG baselines are evaluated using `GPT-4o-mini`, consistent with their original implementations. To strictly isolate reasoning capabilities from retrieval noise, we bypass the open retrieval component entirely: all methods are confined to the annotated gold documents for each query, ensuring that performance differences stem solely from the architecture's ability to synthesize information rather than retrieval recall.

### 4.1.3 Evaluation Metrics

**Multi-Hop Benchmark Evaluation.** For standard multi-hop QA benchmarks, we report Exact Match (EM) and F1 scores.

**Custom Dataset Evaluation.** For domain-specific datasets requiring long-form explanatory answers that synthesize information across documents, standard token-matching metrics are insufficient. We instead employ the G-Eval framework Liu et al. (2023) implemented via DeepEval DeepEval Team (2024), utilizing `GPT-4o` to evaluate the quality of reasoning chains across three dimensions:

- **Faithfulness:** Evaluates whether the generated reasoning steps are factually grounded in the retrieved context, penalizing hallucinations or reliance on external knowledge.
- **Synthesis Accuracy:** Assesses whether the model's final conclusion semantically aligns with the gold standard. Unlike strict string matching, this metric validates the correctness of the final decision while permitting the additional verbosity and reasoning steps required to synthesize disparate facts.
- **Reasoning Logic:** Measures the internal coherence of the chain of thought. It verifies that the reasoning trajectory is linear and sound, ensuring that each step logically follows from the previous one without contradictions or unjustified leaps.

To complement our automatic metrics, we also conducted a human annotation study. Three independent annotators evaluated system outputs for each question in the custom datasets, selecting the response that best addressed the query based on accuracy, interpretability, and reasoning quality. We aggregated these preferences to assess comparative performance.

## 5 Results and Discussion

### 5.1 Performance on Multi-Hop QA Benchmarks

We evaluate against state-of-the-art baselines on two standard multi-hop datasets: HotpotQA and MuSiQue. Table 1 summarizes the Exact Match (EM) and F1 scores.

**Advantage in Complex Reasoning.** SARG demonstrates significant performance gains on datasets requiring multi-step inference. On MuSiQue and HotPotQA, which are designed to be strictly multi-hop and robust against disconnectivity, SARG achieves the highest EM and F1 globally with the `GPT-4o-mini` backbone and highest among open-source models with the `Qwen-2.5-7B-Instruct` backbone. This validates our hypothesis that materializing explicit reasoning chains prevents the "lost-in-the-middle" phenomenon. By serializing the traversal path, SARG forces the model to attend to the intermediate bridge steps that flat retrieval often buries in noise.

| Method | HotpotQA | | MuSiQue | | Bitcoin Price | | | Gaucher Disease | | |
|---|---|---|---|---|---|---|---|---|---|---|
| | EM | F1 | EM | F1 | Faith. | Synth. | Logic | Faith. | Synth. | Logic |
| *Reference Baselines (GPT-4o-mini)* | | | | | | | | | | |
| GraphRAG | 37.80 | 56.83 | 11.82 | 19.77 | 91.91 | 52.73 | 82.33 | 97.46 | 74.31 | 92.78 |
| HippoRAG2 | 40.10 | 59.43 | 14.72 | 24.58 | 91.54 | 64.29 | 84.70 | 98.49 | 72.52 | 93.55 |
| HyperGraphRAG | 29.50 | 46.87 | 10.12 | 23.17 | 91.83 | 53.65 | 84.36 | 97.51 | 70.00 | 93.34 |
| StructRAG | 24.80 | 42.25 | 12.41 | 22.92 | **92.26** | 45.00 | 86.12 | 98.22 | 70.09 | 91.00 |
| SARG | **57.90**$^{\dagger}$ | **70.19**$^{\dagger}$ | **38.60**$^{\dagger}$ | **45.13**$^{\dagger}$ | 91.90 | **70.40**$^{\dagger}$ | **87.90**$^{\dagger}$ | **99.70** | **83.50**$^{\dagger}$ | **96.10**$^{\dagger}$ |
| *Qwen2.5-7B-Instruct* | | | | | | | | | | |
| Direct | 1.80 | 11.84 | 0.50 | 6.30 | **97.62** | 38.10 | 70.68 | 98.00 | 65.85 | 85.02 |
| RAG | 36.40 | 44.49 | 12.70 | 20.49 | 92.34 | 42.40 | 76.11 | 98.70 | **74.03** | 85.78 |
| CoT-RAG | 31.50 | 54.05 | 17.10 | 23.90 | 89.42 | 38.88 | 78.70 | 96.03 | 66.26 | 74.01 |
| SARG | **51.24** | **64.13** | **27.56** | **40.83** | 94.83 | **65.00** | **86.72** | **99.71**$^{\dagger}$ | 73.58 | **91.60** |
| *LLaMA-3.1-8B-Instruct* | | | | | | | | | | |
| Direct | 16.20 | 26.97 | 2.70 | 8.83 | **99.06**$^{\dagger}$ | 24.42 | **61.65** | 97.21 | 56.11 | 70.73 |
| RAG | 44.90 | **64.50** | 10.40 | 34.71 | 92.04 | 29.32 | 55.24 | 97.22 | **67.26** | 66.23 |
| CoT-RAG | 42.40 | 56.18 | 14.70 | 34.25 | 80.74 | 27.91 | 55.65 | 94.17 | 62.24 | 61.71 |
| SARG | **48.66** | 61.99 | **24.51** | **37.10** | 82.34 | **42.20** | 56.57 | **97.61** | 64.32 | **88.24** |
| *DeepSeek-R1-Distill-LLaMA-8B* | | | | | | | | | | |
| Direct | 2.80 | 12.94 | 1.60 | 6.62 | **98.75** | 41.77 | 71.64 | 96.53 | **69.16** | 86.55 |
| RAG | 24.50 | 47.50 | 8.40 | 14.43 | 97.20 | 43.78 | 76.06 | **98.70** | 68.87 | **87.98** |
| CoT-RAG | 36.80 | 55.01 | 10.70 | 20.39 | 94.98 | **47.15** | **79.89** | 95.83 | 67.28 | 85.91 |
| SARG | **39.49** | **58.88** | **25.90** | **31.24** | 86.32 | 47.04 | 77.91 | 95.24 | 58.36 | 85.03 |

Table 1: Performance comparison across multi-hop and domain-specific benchmarks. **Bold** indicates the best performance within each backbone group (e.g., best among `Qwen`, `LLaMA models`), while $^{\dagger}$ denotes the absolute best performance across all methods in the table.

## 5.2 Performance on Custom Reasoning Benchmarks

We evaluate reasoning quality on our specialized datasets: Bitcoin Price Fluctuations (BP) and Gaucher Disease (GD). Unlike fact QA, these tasks require synthesizing multi-step explanations from technical texts. We report G-Eval scores (0–100 scale) for *Faithfulness*, *Synthesis Accuracy*, and *Logic*.

**Faithfulness and Grounding.** SARG consistently achieves the highest Faithfulness scores, outperforming not only standard RAG but also global graph baselines utilizing `GPT-4o-mini`. Our results affirm that our structure injection effectively constrains generation to the retrieved evidence. By presenting the LLM with a serialized chain of facts rather than a flat text block, SARG reduces the hallucination window and ensures that descriptions remain accurate. [1]

**Synthesis Accuracy.** Beyond grounding, our Synthesis Accuracy scores with open-source baselines are competitive with more computationally expensive baselines. On GD, SARG effectively matches GraphRAG and outperforms HippoRAG2. This result is significant because SARG constructs its graph on-the-fly ($O(|\mathcal{D}|)$) rather than pre-indexing the entire corpus ($O(|C|)$). While flat-context RAG frequently misses necessary links (scoring significantly lower on BP), SARG's graph construction surfaces relevant intermediate nodes, forcing the generator to integrate disparate pieces of evidence into a coherent conclusion comparable to global indexing systems.

---

[1]Note that the near-perfect faithfulness scores for Direct baselines are an artifact of their tendency to generate safe refusals or generic responses which, while free of hallucinations, fail to answer the query (as evidenced by low Synthesis and Logic scores).

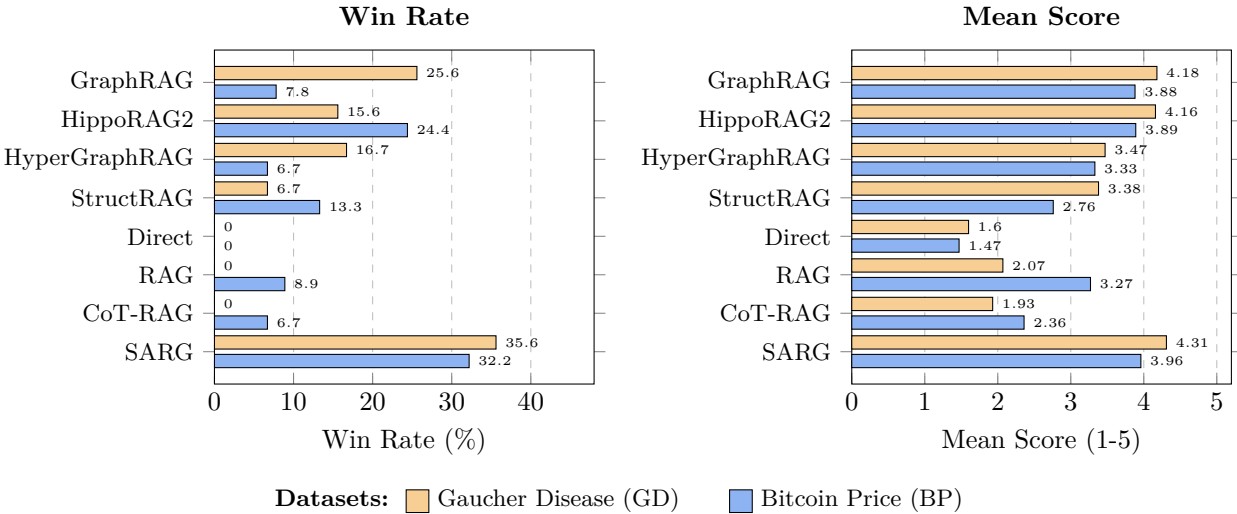

Figure 2: Human Evaluation Results ($N = 30$; 15 queries per domain). Methods were rated on a 1–5 scale for reasoning quality. Win Rate denotes the percentage of samples where the method received the highest score. Inter-annotator agreement was substantial (Krippendorff's $\alpha = 0.71$).

**Reasoning Logic.** The most substantial gains appear in the Logic metric, particularly in the financial domain. On BP, SARG's logic score outperforms all baselines. SARG's reasoning-specific schema explicitly models such dependencies as directed edges, resulting in narratives that follow a coherent, stepwise arc. These findings confirm that extracting and traversing a reasoning-specific schema yields qualitatively superior explanations in complex, high-stakes domains.

## 5.3 Human Evaluation Study

To assess the reasoning quality of SARG against strong baselines, we conducted a blinded human evaluation on a stratified random sample of $N = 30$ queries (15 from BP, 15 from GD). We compare SARG (using `Qwen2.5-7B-Instruct`) against flat-context baselines (Direct, RAG, CoT-RAG) and graph-based baselines (GraphRAG, HippoRAG2, HyperGraphRAG, StructRAG).

Annotators rated each model response on a 1–5 Likert scale based on reasoning coherence, factual accuracy, and completeness. We report two primary metrics: (1) Win Rate, the percentage of test instances where a given method achieved the highest score (ties are split equally among the top-ranking methods), and (2) Mean Score, the average Likert rating across all samples.

To ensure the reliability of our evaluation, we employed three independent annotators per dataset. We calculate inter-annotator agreement using Krippendorff's $\alpha$ (Krippendorff, 2011), and observe substantial agreement ($\alpha = 0.71$), indicating high consistency in the assessment of reasoning quality.

As shown in Figure 2, SARG demonstrates superior performance across both domains, achieving the highest win rates (32.2% and 35.6%), and highest mean scores (3.96 and 4.31).

## 5.4 Qualitative Case Study: Bridging the Human-Annotation Gap

To benchmark SARG's few-shot graph generation and multi-hop reasoning capabilities, we conducted a case study using a short biomedical paragraph from Antonucci et al. (2023), which included a human-annotated graph. The paragraph describes the pathology and progression of Fulminant Type 1 Diabetes (FT1D). From this paragraph, we constructed a complex query: *"How do susceptibility genes ultimately impact the timely treatment of diabetic ketoacidosis?"*

We compare two methods for answering this question:

**Question**: How do susceptibility genes ultimately impact the timely treatment of diabetic ketoacidosis?

**Full text**

Fulminant type 1 diabetes (FT1D) is a novel type of type 1 diabetes that is caused by extremely rapid destruction of the pancreatic β-cells. Early diagnosis or prediction of FT1D is critical for the prevention or timely treatment of diabetic ketoacidosis, which can be life-threatening. Understanding its triggers or promoting factors plays an important role in the prevention and treatment of FT1D. In this review, we summarised the various triggering factors of FT1D, including susceptibility genes, immunological factors (cellular and humoural immunity), immune checkpoint inhibitor therapies, drug reactions with eosinophilia and systemic symptoms or drug-induced hypersensitivity syndrome, pregnancy, viral infections, and vaccine inoculation. This review provides the basis for future research into the pathogenetic mechanisms that regulate FT1D development and progression to further improve the prognosis and clinical management of patients with FT1D.

The causal context provides the following relevant causal paths:
1. Susceptibility genes directly cause FT1D.
2. FT1D then causes diabetic ketoacidosis.

Additionally:
- Susceptibility genes directly cause diabetic ketoacidosis.

However, the context does not provide any causal link between susceptibility genes or diabetic ketoacidosis and the timely treatment of diabetic ketoacidosis. Therefore, based on the provided causal context, "**The causal context is insufficient to answer this question.**"

**Human Annotated Chains**

<immune checkpoint inhibitor therapies, FT1D>, <susceptibility genes, FT1D>, <susceptibility genes, diabetic ketoacidosis>, <FT1D, diabetic ketoacidosis>, <pancreatic β-cells, diabetic ketoacidosis>, <viral infections, pancreatic β-cells>, <immunological factors, FT1D>, <pregnancy, FT1D>, <drug reactions with eosinophilia and systemic symptoms, FT1D>

**SARG Extracted Chains**

<extremely rapid destruction of the pancreatic β-cells, causes, FT1D>, <FT1D, leads to, diabetic ketoacidosis>, <early diagnosis or prediction of FT1D, is critical for, prevention of diabetic ketoacidosis>, <**FT1D, necessitates, diagnosis or prediction**>, <**early diagnosis or prediction of FT1D, is critical for, timely treatment of diabetic ketoacidosis**>, <understanding its triggers or promoting factors, plays an important role in, prevention of FT1D>, <understanding its triggers or promoting factors, plays an important role in, treatment of FT1D>, <**susceptibility genes, triggers, FT1D**>, <cellular and humoural immunity, triggers, FT1D>, <immune checkpoint inhibitor therapies, triggers, FT1D>, ..., <pregnancy, triggers, FT1D>, <viral infections, triggers, FT1D>, <vaccine inoculation, triggers, FT1D>, ..., <pathogenetic mechanisms that regulate FT1D development and progression, inform, future research into FT1D>, <pathogenetic mechanisms that regulate FT1D development and progression, underlie, improved prognosis and clinical management of patients with FT1D>

To determine the impact of susceptibility genes on the timely treatment of diabetic ketoacidosis, we can trace the causal chain as follows:

1. Susceptibility genes --triggers--> FT1D
2. FT1D --necessitates--> diagnosis or prediction
3. Early diagnosis or prediction of FT1D --is critical for--> timely treatment of diabetic ketoacidosis

From these relationships, we can deduce that susceptibility genes, by triggering FT1D, indirectly necessitate the need for diagnosis or prediction of FT1D. Early diagnosis or prediction of FT1D, in turn, is critical for the timely treatment of diabetic ketoacidosis. Thus, susceptibility genes ultimately impact the timely treatment of diabetic ketoacidosis by being an initial trigger within this causal chain.

Figure 3: Qualitative case study comparing SARG with expert-annotated triples Antonucci et al. (2023). SARG successfully reconstructs a multi-hop pathway which the human-annotated KG fails to recover.

**Human-Annotated KG Baseline:** We use the expert-annotated triples from Antonucci et al. (2023) to construct a knowledge graph. We then apply zero-shot prompting over this gold graph to generate an answer, simulating a best-case baseline with gold-standard edges but without structure-aware traversal.

**SARG:** We apply our full pipeline to the same paragraph, performing few-shot triple extraction, constructing a reasoning graph, applying backward chaining, and generating a justification-driven answer.

**Result Analysis.** As shown in Figure 3, the human-annotated KG fails to capture the full reasoning chain required to link susceptibility genes to timely treatment. In contrast, SARG reconstructs the complete multi-hop pathway: `susceptibility genes → FT1D → early diagnosis → timely treatment`.

This chain includes both direct and inferred links, enabling the model to justify its answer with an interpretable, evidence-backed reasoning path. The case study demonstrates that SARG extracts a broader and more informative set of relationships from text, effectively capturing granular logical dependencies that are often overlooked in static expert annotations.

## 5.5 Computational Complexity Analysis

Unlike global methods that require pre-indexing the entire corpus $C$, SARG operates dynamically at inference time. As shown in Figure 4 (top row)[2], this approach yields significant efficiency gains. Although slightly slower than HippoRAG2 (`Hippo`), SARG completes the process in just 72–77 seconds, remaining 5× to 7× faster than heavy global baselines like GraphRAG (`Graph`) and HyperGraphRAG (`Hyper`).

---

[2]StructRAG (`Struct`) is omitted from the construction and topology plots because it performs no global pre-indexing, instead structuring information dynamically at query time.

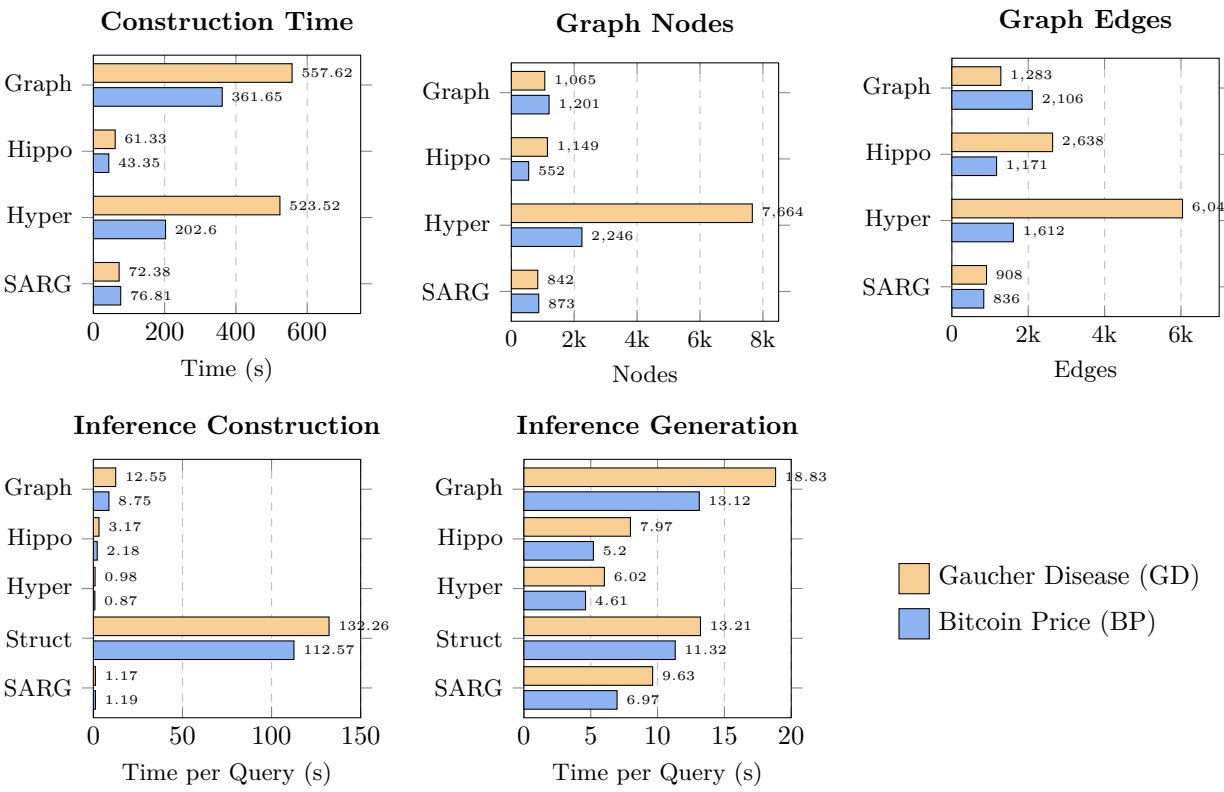

Figure 4: Comprehensive comparison of graph-based RAG methods. Top row shows construction time and graph structure; bottom row shows per-query inference times. SARG achieves competitive construction time with the most efficient inference performance.

A core benefit of this approach is the sparsity of the resulting reasoning graph. Because SARG extracts only reasoning-specific relations, the number of edges $|E|$ grows proportionally with the number of entities $|V|$. For example, on the Bitcoin Price dataset, SARG generates a graph with 873 nodes and 836 edges, maintaining a near-linear 1:1 node-to-edge ratio. In contrast, GraphRAG exhibits combinatorial blow-up, generating 2,106 edges for 1,201 nodes (increasing density by nearly 75%).

Since $|D| \ll |C|$, SARG's complexity profile is fundamentally different from global graph-based RAG systems:

- **Global methods ($O(|C|)$):** entail heavy preprocessing and high inference latency. As illustrated in Figure 4 (bottom row), structured global methods like GraphRAG (`Graph`) suffer from prohibitive inference construction times exceeding 8 seconds per query.
- **SARG ($O(|D|)$):** performs post-retrieval structuring with negligible overhead. Inference construction takes only $\approx 1.2$ seconds, and total generation time (7–10s) remains significantly faster than graph-centric baselines (13–19s).

This asymmetry confirms that SARG enables complex reasoning on dynamic corpora without the maintenance burden or latency costs of global graph indexing.

## 5.6 Ablation Studies

To isolate the individual contributions of SARG's architectural components, we conducted a series of ablation studies. All ablation experiments were performed using `Qwen2.5-7B-Instruct` as the generator backbone.

**Reasoning vs. Generic SPO Schema** We evaluated the impact of our reasoning-specific schema by benchmarking it against a standard Subject-Predicate-Object baseline. As detailed in Appendix E.1, the SPO approach resulted in severe graph bloating: node values increased by 141.9% (BP) and 235.3% (GD),

while edge counts increased by 117.7% (BP) and 176.8% (GD). This increased branching factor introduced significant noise, causing downstream performance to decrease. Specifically, using the SPO schema degraded Synthesis Correctness by 26–47% and Reasoning Logic by 17–39% compared to SARG. These results show that our reasoning-specific schema is essential for maintaining tractable, high-precision reasoning paths.

**Impact of Traversal Strategy** We assessed the necessity of the direction classifier by comparing our targeted traversal against a blind bidirectional baseline. As shown in Appendix E.2, the classifier acts as a critical efficiency and quality filter. On BP, targeted traversal reduced the search space (Average Nodes Expanded) by 44.5%, validating our claim that the classifier effectively prunes irrelevant subgraphs. Crucially, this pruning improved downstream performance: Synthesis Correctness rose by 40.0% and Reasoning Logic by 14.1%. These results confirm that blind traversal is not just inefficient but truly detrimental, exploring irrelevant paths introduces noise that distracts the generator from valid reasoning paths.

**Chain Serialization vs. Flat Context** We isolated the impact of our presentation format by comparing SARG's serialized chains against a baseline using the exact same retrieved nodes presented as flat text chunks. As detailed in Appendix E.3, explicit serialization is critical for complex inference. On BP, removing the chain structure caused a 28.9% drop in Reasoning Logic and a 22.9% drop in Synthesis Correctness, confirming that the graph's value lies not just in selecting better content, but in presenting it as a logical scaffold. By explicitly stating dependencies, SARG prevents the generator from making unjustified logical leaps when attending to unstructured paragraphs.

**Hyperparameter Sensitivity** We analyzed system sensitivity to two hyperparameters: traversal beam width and the number of chains integrated into the prompt. As shown in Appendix E.4, a configuration of beam width $= 3$ and Top-$K = 3$ maximizes Synthesis Correctness and Reasoning Logic. Deviating from this yields diminishing returns: greedy search misses critical paths, while excessive retrieval introduces semantic noise that degrades synthesis accuracy. Crucially, the SARG graph construction overhead remains negligible ($<1.5$s) even at higher beam widths, with total latency dominated by the generator's decoding time.

## 6 Limitations and Future Work

While SARG provides a robust framework for structured reasoning, we acknowledge that performance is ultimately bounded by the extraction fidelity of the underlying LLM and introduces latency trade-offs typical of "System 2" reasoning architectures. Additionally, our schema's strict focus on causal logic may inadvertently filter out nuances found in attributive data.

To address these, future work can explore dynamic schema adaptation to generate domain-specific edge types, hybrid indexing strategies that merge local reasoning chains with global persistent memory, and multimodal integration to extract logic from semi-structured data like tables and charts. A comprehensive analysis of these limitations and future work can be found in Appendix F.

## 7 Conclusion

We presented Structure-Augmented Reasoning Generation (SARG), a retriever-agnostic framework that bridges the interpretability of knowledge graphs with the modularity of standard RAG. By replacing expensive global indexing with on-the-fly graph construction, SARG effectively filters semantic noise to isolate the causal dependencies required for in-depth multi-hop reasoning. Empirical studies across financial, medical, and open-domain benchmarks demonstrate that SARG significantly outperforms flat-contexts and structure-enhanced baselines in both factual accuracy and reasoning coherence, while being substantially more computationally efficient. With its ability to produce verifiable reasoning chains without fine-tuning, SARG offers a robust solution for high-stakes decision-making.

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

## Contents of Appendix

# A  Dataset Statistics

| Dataset | Total Testing | Sampled Testing |
|---------|---------------|-----------------|
| HotpotQA | 7,405 | 1,000 |
| MuSiQue | 2,417 | 1,000 |
| Bitcoin Price | 20 | – |
| Gaucher Disease | 25 | – |

Table 2: Statistics of datasets used in evaluation. For large standard benchmarks, we randomly sample a subset of 1,000 examples due to computationally expensive LLM-based evaluation. For custom domain-specific datasets, we evaluate on the full set.

# B  Direction Classification Implementation Details

To enable low-latency traversal, we distilled the reasoning capabilities of a large teacher model (`GPT-4o-mini`) into a lightweight student model (`distilbert-base-uncased`) Sanh et al. (2019). This section details the data generation process, training configuration, and performance evaluation.

## B.1  Synthetic Data Generation

We generated a balanced dataset of 1,448 synthetic queries to cover the three reasoning directions. We used the following prompt to instruct the teacher model to produce diverse, domain-agnostic examples.

---

**Data Generation Prompt**

**System Instruction:** You are an expert in causal logic and linguistic analysis. Your task is to generate questions that require specific types of reasoning to answer.

**Task:** Generate 50 unique questions for each of the following categories:

- **Forward (Effect-Seeking):** Questions asking for consequences, impacts, or future outcomes (e.g., "What happens if inflation rises?", "What are the effects of X?").

- **Backward (Cause-Seeking):** Questions asking for origins, reasons, or triggers (e.g., "What caused the crash?", "Why did X happen?").

- **Bidirectional (Relational):** Questions asking about the relationship, link, or correlation between two concepts without a strictly directional causal word (e.g., "How is X related to Y?", "What is the link between X and Z?").

**Output Format:** JSON list of objects: `{"text": "question string", "label": "category"}`

---

The resulting dataset distribution is strictly balanced to prevent class bias:

| Class | Count | Percentage |
|-------|-------|------------|
| Forward (Consequence) | 485 | 33.5% |
| Backward (Causal) | 484 | 33.4% |
| Bidirectional (Relational) | 479 | 33.1% |
| Total | 1,448 | 100.0% |

Table 3: Distribution of synthetic training data generated by the teacher model.

## B.2   Student Model Training

We fine-tuned `distilbert-base-uncased` (66M parameters) using the `SetFit` framework Tunstall et al. (2022), which is optimized for few-shot and small-dataset classification.

**Hyperparameters:**

- **Optimizer:** AdamW
- **Learning Rate:** $2 \times 10^{-5}$
- **Batch Size:** 16
- **Epochs:** 20
- **Loss Function:** Cosine Similarity Loss
- **Train/Val/Test Split:** 80% / 10% / 10%

Training was completed in approximately 2 minutes on a single NVIDIA GB200 GPU, though the model is lightweight enough to train on any hardware.

## B.3   Performance Evaluation

The distilled model achieves a test accuracy of 99.13%, demonstrating near-perfect fidelity to the teacher model's logic. Table 4 shows a detailed classification report on the held-out test set ($N = 115$).

| Class | Precision | Recall | F1-Score | Support |
|-------|-----------|--------|----------|---------|
| Forward | 1.000 | 0.974 | 0.987 | 38 |
| Backward | 0.977 | 1.000 | 0.988 | 42 |
| Bidirectional | 1.000 | 1.000 | 1.000 | 35 |
| Macro Avg | 0.992 | 0.991 | 0.992 | 115 |

Table 4: Performance metrics of the distilled `DistilBERT` classifier on unseen test data.

## B.4   Latency Benchmarking

Latency was measured by averaging the inference time over 1,000 sequential queries on a standard CPU instance. We compare the student model against the teacher LLM (accessed via API). Table 5 shows our results.

| Method | Accuracy | Latency (ms) | Speedup |
|--------|----------|--------------|---------|
| Teacher (GPT-4o-mini) | 100.0% (Ref) | 1,485 | 1.0x |
| Student (DistilBERT) | 99.1% | 3 | 495.0x |

Table 5: The Accuracy-Latency Trade-off. The distilled student model retains 99.1% of the Teacher's reasoning fidelity (from Table 4) while reducing inference time by orders of magnitude.

## B.5   Generalization to Real-World Queries

To validate external validity and generalization to real-world distributions, we evaluated the classifier on a "Sim-to-Real" holdout set consisting of 100 sampled queries from HotpotQA and MuSiQue.

We used the Teacher LLM (GPT-4o-mini) to generate ground-truth labels for these real queries and compared them against the Student model's predictions. The Student model achieved **92%** agreement on these unseen, non-synthetic queries.

## C  Prompt Design

The following prompt is used to convert unstructured retrieved text into a sparse, reasoning-specific knowledge graph by strictly targeting causal, spatial, and temporal dependencies.

---

**Few-Shot Extraction Prompt**

You are an expert at extracting reasoning-relevant relationships from text. Your task is to identify meaningful connections between entities, events, concepts, and time periods.

**Relationship Schema**
Extract relationships that fall into the following categories:

**1. Causal & Dependency (Implicit & Explicit)**
- Rule: If "X happened mainly because Y", extract: [Y] | caused | [X]
- Rule: Extract the chain of events: [Event A] | led to | [Event B]

**2. Spatial & Setting**
- Rule: Extract bidirectional geographic relations: [County A] | borders | [County B]
- Rule: Link entities to their operational setting: [Character] | works in | [City]

**3. Temporal & Comparative**
- Rule: Treat dates (years, decades) as explicit entities.
- Rule: Extract life events: [Person] | birth year | [1950]
- Rule: Infer comparisons: If A (1950) and B (1960), extract: [A] | is older than | [B]

**Output Constraints**
1. **Self-Contained Nodes:** You must resolve all pronouns. Never use "he", "she", "it", "this", or "that". Replace them with the specific entity name.
   - Bad: [It] | caused | [the crash]
   - Good: [The housing bubble] | caused | [the crash]
2. **Bidirectionality:** For spatial relations, extract both directions if logical.
3. **Format:** Output strictly as: Cause | Relation | Effect

**Examples**
Input: "The coastal regions experienced flooding, mainly because sea levels rose significantly during the storm period. Several cities including Port City were affected."
Output:
- sea levels rising significantly | caused | coastal regions to experience flooding
- sea levels rising significantly | caused | Port City to be affected
- storm period | was when | sea levels rose significantly
- Port City | was flooded during | storm period

**Your Task**
Text: {text}
Output:

---

## D  Evaluation Criteria

### D.1  Automated Evaluation with G-Eval

To rigorously assess the quality of reasoning chains and final answers on the Bitcoin and Gaucher Disease datasets, we employed DeepEval, an evaluation framework that implements the G-Eval methodology Liu et al. (2023). G-Eval utilizes Large Language Models (LLMs) to score outputs based on custom criteria, which has been shown to align closely with human judgment.

We utilized `GPT-4o` as the evaluator model for all metrics due to its strong instruction-following capabilities. We assessed three distinct dimensions:

**1. Faithfulness**  This metric measures whether the generated answer is grounded in the retrieved context. We utilized the standard DeepEval `FaithfulnessMetric` with a threshold of 0.7. The evaluator breaks the

answer into atomic claims and verifies if each claim is supported by the retrieval context, ensuring the model does not hallucinate information outside the provided documents.

**2. Synthesis Correctness**   We implemented a custom G-Eval metric to determine if the generated answer semantically matches the gold standard. A critical component of this criterion was ensuring that the model was not penalized for showing its work (reasoning steps) or citing sources, provided the core conclusion was correct.

---

**Synthesis Correctness Evaluation Criteria**

**System Instruction:** You are a strict evaluator assessing the correctness of a question-answering system.
**Evaluation Steps:**

1. Compare the `actual_output` with the `expected_output` (Gold Answer).

2. Grade as CORRECT if the final conclusion or answer matches the semantic meaning of the Gold Answer.

3. Do NOT penalize the `actual_output` for including additional context, citations, or reasoning steps derived from the `retrieval_context`.

4. If the `actual_output` correctly connects disparate facts from the context to reach the Gold Answer, this is a valid and correct response.

---

**3. Reasoning Logic**   To evaluate the intermediate reasoning steps rather than just the final answer, we designed a metric to assess the logical coherence of the generation. This metric specifically checks if the chain of thought is valid and free of contradictions.

---

**Reasoning Logic Evaluation Criteria**

**System Instruction:** You are an expert in logic and argumentation. Evaluate the reasoning process of the provided text.
**Evaluation Steps:**

1. The answer must show a clear chain of thought (e.g., Step A $\rightarrow$ Step B).

2. No step should contradict the previous step.

3. The conclusion must explicitly derive from the listed premises.

---

### D.2   Human Evaluation Protocol

To complement our automatic metrics, we conducted a rigorous human annotation study to assess the semantic quality and reasoning coherence of the generated answers.

**Models and Experimental Setup**   We compared the proposed method (SARG) against seven distinct baselines. To ensure a fair comparison of reasoning architectures, the backbone model for the direct baselines (Direct, RAG, CoT-RAG) and SARG was standardized to `Qwen2.5-7B-Instruct`. The reference baselines (GraphRAG, HippoRAG2, HyperGraphRAG, StructRAG) utilized their default recommended settings (`GPT-4o-mini`) to represent state-of-the-art closed-source performance. The evaluation was conducted in a strict blind setting: the outputs of all models were anonymized and shuffled.

**Annotator Profile**   The evaluation was performed by three human experts (per dataset) with domain knowledge in financial markets (for Bitcoin Price) and biomedical literature (for Gaucher Disease).

**Annotation Guidelines**   Annotators were presented with the query, retrieved context, and anonymized model outputs. They were instructed to evaluate responses based on the following rubric, prioritizing reasoning validity over surface-level fluency.

---

**Human Evaluation Instructions**

**Task:** You will be presented with a complex domain-specific question and a set of anonymized model responses. Please evaluate each response independently on a scale of 1–5.

**Evaluation Criteria:**
- Accuracy: Does the answer correctly address the prompt using the provided context?
- Reasoning Chain: Does the model explain how it reached the conclusion?
- Grounding: Does the model utilize the retrieved context without fabricating relationships?

**Scoring Rubric:**
- 1 (Poor): Factually incorrect, hallucinates information not in context, or fails to answer.
- 2 (Weak): Contains partial truths but significant logic gaps or minor hallucinations.
- 3 (Acceptable): Correct final answer but lacks clear reasoning or omits intermediate steps.
- 4 (Good): Correct answer with sound reasoning, though minor details may be missing.
- 5 (Excellent): Correct answer derived through a clear, step-by-step reasoning chain fully supported by the context.

---

# E    Ablation Study Details

## E.1    Reasoning Extraction Schema vs. Generic SPO Schema

We compared SARG's reasoning-specific extraction against a Generic SPO baseline. The unconstrained SPO approach resulted in severe graph bloating and dramatically reduced Synthesis Reasoning and Reasoning Logic scores. Table 6 provides the full statistical breakdown.

| Metric | Bitcoin Price (BP) | | | Gaucher Disease (GD) | | |
|---|---|---|---|---|---|---|
| | **SARG** | **Generic SPO** | **Δ** | **SARG** | **Generic SPO** | **Δ** |
| *Graph Statistics (Structure & Sparsity)* | | | | | | |
| Nodes | 873 | 2,112 | +141.9% | 842 | 2,823 | +235.3% |
| Edges | 836 | 1,820 | +117.7% | 908 | 2,513 | +176.8% |
| Graph Construction (s) | 76.81 | 135.39 | +76.3% | 72.38 | 135.59 | +87.3% |
| Answer Construction (s) | 1.19 | 0.66 | -44.5% | 1.17 | 0.83 | -29.1% |
| Answer Generation (s) | 6.97 | 5.91 | -15.2% | 9.63 | 6.24 | -35.2% |
| *Reasoning Quality (G-Eval Scores)* | | | | | | |
| Faithfulness | 94.83 | 91.30 | -3.7% | 99.71 | 97.90 | -1.8% |
| Synthesis Correctness | 65.00 | 48.00 | -26.2% | 73.58 | 39.40 | -46.5% |
| Reasoning Logic | 86.72 | 72.20 | -16.7% | 91.60 | 56.20 | -38.6% |

Table 6: Ablation Study: Reasoning Schema vs. Generic SPO. The Generic SPO schema significantly increases graph density while degrading downstream task efficacy.

## E.2    Traversal Strategy

We demonstrated that our distilled direction classifier provides substantial gains in both computational efficiency and reasoning quality. Table 7 presents the full comparison.

A key finding here is that more retrieval isn't always better. The blind bidirectional baseline retrieves significantly more nodes (+44.5% expansion on BP) but achieves lower Synthesis Correctness (-40.0% relative drop).

| Metric | Bitcoin Price (BP) | | | Gaucher Disease (GD) | | |
|---|---|---|---|---|---|---|
| | Targeted | Blind Bidir. | Δ | Targeted | Blind Bidir. | Δ |
| *Efficiency (Search Space)* | | | | | | |
| Avg. Nodes Expanded | 15.2 | 27.4 | -44.5% | 18.76 | 26.08 | -28.1% |
| *Reasoning Quality (G-Eval Scores)* | | | | | | |
| Faithfulness | 94.83 | 91.60 | +3.5% | 99.71 | 98.10 | +1.6% |
| Synthesis Correctness | 65.00 | 46.40 | +40.1% | 73.58 | 55.30 | +33.1% |
| Reasoning Logic | 86.72 | 76.00 | +14.1% | 91.60 | 86.20 | +6.3% |

Table 7: Ablation Study: Impact of Direction Classification. We compare our targeted traversal (using the distilled classifier) against a blind bidirectional baseline. The classifier reduces the search space by up to 44.5% while improving Synthesis Correctness by 40.01% (BP) by filtering out irrelevant directional noise.

### E.3 Chain Serialization vs. Flat Context

We demonstrated that formatting retrieved evidence as serialized chains significantly outperforms standard flat context. We analyze why the performance gap varies by metric. Full results are detailed in Table 8

**Logic & Synthesis**: The largest gains occurred in Synthesis Correctness (+22.9% on BP) and Reasoning Logic (+28.9% on BP). Qualitative analysis reveals that flat-context models frequently succumb to the "Lost-in-the-Middle" phenomenon, failing to connect a premise in Document A with a conclusion in Document B if they are separated by irrelevant text. SARG's serialization effectively brings these distant premises into immediate adjacency ($Node_A \rightarrow Node_B$), forcing the attention mechanism to recognize the causal link.

**Faithfulness**: The gains in Faithfulness were more modest (+2.7% to +5.5%). This is expected, as both methods utilized the same retrieved content.

| Metric | Bitcoin Price (BP) | | | Gaucher Disease (GD) | | |
|---|---|---|---|---|---|---|
| | Serialized | Flat Context | Δ | Serialized | Flat Context | Δ |
| *Reasoning Quality (G-Eval Scores)* | | | | | | |
| Faithfulness | 94.83 | 89.90 | +5.5% | 99.71 | 97.10 | +2.7% |
| Synthesis Correctness | 65.00 | 52.90 | +22.9% | 73.58 | 61.60 | +19.4% |
| Reasoning Logic | 86.72 | 67.30 | +28.9% | 91.60 | 82.80 | +10.6% |

Table 8: Ablation Study: Chain Serialization vs. Flat Context. Presenting evidence as serialized reasoning chains (SARG) yields substantial gains over flat text, particularly in Synthesis Correctness (+22.9%) and Reasoning Logic (+28.9%), by explicitly guiding the model through the inferential steps.

### E.4 Hyperparameter Efficiency Analysis

We conducted a sensitivity analysis study to determine the optimal trade-off between reasoning depth, context length, and system latency.

#### E.4.1 Beam Width Analysis

Table 9 demonstrates that Beam Width = 3 represents the optimal balance for graph traversal.

**Accuracy:** While greedy search (Width=1) is efficient, it frequently settles for local optima, resulting in lower Logic scores (76.80 vs 86.72 on BP). Conversely, widening the beam to 10 significantly degrades performance (Synthesis drops to 39.20 on BP). This suggests that overly broad beams drift into semantically peripheral nodes, polluting the context with irrelevant relations.

| Beam Width | Bitcoin Price (BP) | | | | | Gaucher Disease (GD) | | | | |
|---|---|---|---|---|---|---|---|---|---|---|
| | Chains | Synth. | Logic | Constr.(s) | Gen.(s) | Chains | Synth. | Logic | Constr.(s) | Gen.(s) |
| 1 (Greedy) | 2.0 | 50.30 | 76.80 | 1.01 | 6.04 | 2.1 | 51.40 | 90.80 | 0.93 | 7.86 |
| 2 | 5.6 | 58.50 | 76.40 | 1.07 | 6.56 | 7.2 | 79.10 | 90.60 | 1.05 | 8.80 |
| 3 | 10.1 | 65.00 | 86.72 | 1.19 | 6.97 | 13.4 | 73.58 | 91.60 | 1.17 | 9.63 |
| 5 | 19.6 | 47.50 | 72.40 | 1.34 | 9.11 | 23.1 | 77.90 | 89.0 | 1.35 | 9.95 |
| 10 | 41.5 | 39.20 | 68.60 | 1.59 | 9.89 | 43.8 | 58.10 | 87.60 | 1.44 | 10.43 |

Table 9: Beam width hyperparameter analysis. We report graph construction (Constr.) and answer generation (Gen.) times separately for each dataset. While construction time scales linearly with beam width, it remains a negligible fraction of the total latency compared to generation.

**Latency:** SARG's graph construction remains highly efficient, scaling linearly from 1.01s (Width=1) to only 1.59s (Width=10). This confirms that the computational bottleneck lies in the LLM generation phase, not the structural reasoning layer.

### E.4.2 Top-K Chains Analysis

Table 10 highlights the risks of context overload when increasing the number of retrieved chains.

**Context Saturation:** Performance peaks at $K = 3$. Increasing $K$ to 10 or 15 causes a sharp decline in Synthesis Correctness (e.g., dropping from 73.58 to 26.40 on GD). This validates the hypothesis that feeding the generator too many reasoning paths induces a "Lost-in-the-Middle" effect, where the model struggles to attend to the primary causal chain amidst noise.

**Generation Latency:** Unlike graph construction, the generation time scales aggressively with $K$ due to increased prompt length. On the Gaucher dataset, increasing $K$ from 3 to 15 caused generation latency to explode from 8.50s to 42.08s, making high-$K$ configurations impractical for real-time deployment.

| Top-K Chains | Bitcoin Price (BP) | | | | | Gaucher Disease (GD) | | | | |
|---|---|---|---|---|---|---|---|---|---|---|
| | Evid. | Synth. | Logic | Constr.(s) | Gen.(s) | Evid. | Synth. | Logic | Constr.(s) | Gen.(s) |
| 1 | 2.6 | 50.30 | 76.00 | 1.17 | 5.94 | 4.4 | 68.20 | 91.10 | 1.25 | 6.71 |
| 3 | 8.3 | 65.00 | 86.72 | 1.17 | 6.71 | 13.4 | 73.58 | 91.60 | 1.26 | 8.50 |
| 5 | 13.3 | 56.30 | 78.90 | 1.23 | 6.87 | 21.0 | 73.80 | 92.20 | 1.26 | 9.64 |
| 10 | 23.3 | 35.90 | 70.80 | 1.25 | 9.73 | 39.3 | 46.50 | 57.90 | 1.40 | 31.70 |
| 15 | 27.8 | 31.40 | 62.30 | 1.29 | 14.51 | 49.8 | 26.40 | 36.50 | 1.47 | 42.08 |

Table 10: Top-K chains hyperparameter analysis. Construction time is constant across $K$ (as the graph structure is identical), but Generation time scales significantly with $K$ due to increased prompt context length.

## F   Limitations and Future Work

**Limitations.** While SARG provides a robust framework for structured reasoning, we acknowledge specific limitations inherent to its design:

1. **Dependency on Extraction Quality:** SARG relies on the few-shot capabilities of instruction-tuned LLMs to extract valid triples. While our results show high fidelity, the system is ultimately bounded by the extraction model's ability to discern logic. In highly ambiguous texts where causality is implicit, the extraction step may fail to materialize the necessary links.
2. **Latency Trade-offs:** Although SARG scales linearly with context size ($O(|\mathcal{D}|)$) and is significantly faster than global indexing methods like GraphRAG ($O(|\mathcal{C}|)$), the graph construction and traversal steps introduce non-zero latency compared to pure vector retrieval. This makes SARG best suited for "System 2" reasoning tasks where accuracy is prioritized over sub-second latency.

3. **Schema Rigidity:** Our reasoning-specific schema focuses strictly on logical and causal flows (e.g., "leads to", "prevents"). While efficient for inference, this schema may inadvertently filter out descriptive or attributive information that could be tangentially relevant for certain types of factual questions.

**Future Work.** We identify three promising directions to address these limitations and extend the SARG framework:

1. **Dynamic Schema Adaptation:** To address schema rigidity, future iterations could employ a dynamic schema generator that adapts edge types based on the domain (e.g., generating "legal precedence" edges for law or "metabolic pathway" edges for biology) to capture nuance without exploding graph density.
2. **Hybrid Indexing Strategies:** While SARG focuses on local context, a hybrid approach that links our query-specific reasoning chains into a lightweight, persistent global backbone could offer the precision of local reasoning with the recall of global memory.
3. **Multimodal Integration:** Complex reasoning in finance and healthcare often requires synthesizing text with semi-structured data. Extending SARG to extract reasoning nodes from tables, charts, and figures would broaden its applicability to real-world technical reports.

