# OpenReview forum: "Structure-Augmented Reasoning Generation"
_TMLR — Under review for TMLR_

### Review · Reviewer_vf5P · 2026-03-01

**Summary Of Contributions:**

The paper proposes Structure-Augmented Reasoning Generation (SARG), a modular, post-retrieval framework designed to address the knowledge synthesis bottleneck in standard Retrieval-Augmented Generation (RAG). Instead of relying on expensive, corpus-wide global graph indexing, SARG dynamically constructs a local reasoning graph from retrieved documents by extracting reasoning-specific (cause, relation, effect) triples. It then utilizes a lightweight, distilled direction classifier to guide a semantic beam-search traversal, surfacing multi-hop reasoning chains that are serialized and integrated into the generator LLM's prompt.

***Key Strengths:***

- Architectural Novelty: The post-retrieval, on-demand graph construction preserves RAG's flexibility while offering the interpretability of graph-based reasoning.

- Design Choices: Enforcing a reasoning-specific schema controls graph noise compared to generic OpenIE, and the use of a distilled direction classifier is a smart systems optimization to control branching without incurring repeated LLM latency costs.

- Interpretability: By serializing explicit reasoning chains, the method provides traceable, step-by-step justifications for its answers.

***Key Weaknesses:***

- Experimental Confounds: Comparing methods using different backbone models and artificially restricting the context to "gold documents" undermines the fairness of the comparisons against global graph baselines.

- Missing Intermediate Evaluations: The paper lacks quantitative evaluation of the underlying LLM-based triple extraction fidelity and entity resolution accuracy.

- Evaluation Robustness: Heavy reliance on LLM-as-a-judge for very small custom datasets introduces risks of bias and overfitting.

**Audience:**

Yes

**Audience Explanation:**

Addressing the "knowledge synthesis" and multi-hop reasoning bottlenecks in RAG is a highly active and important area of research. SARG’s plug-and-play, post-retrieval design speaks directly to real-world deployment constraints—specifically the need to handle dynamic corpora without the prohibitive latency and maintenance costs of global graph indexing (e.g., GraphRAG). Furthermore, the framework's emphasis on producing verifiable, serialized reasoning chains will be of strong interest to practitioners and researchers working on trustworthy AI in high-stakes domains like finance, law, and biomedicine.

**Broader Impact Concerns:**

The paper focuses on improving automated reasoning in complex, high-stakes domains such as biomedical pathology and financial markets. A key risk of the SARG approach is the potential over-interpretation of noisy, LLM-extracted associative relations as definitive "causal" links. Because SARG explicitly serializes these extracted triples into structured reasoning chains, it risks producing highly convincing, well-structured justifications that are built on fabricated or hallucinated dependencies. In medical or financial contexts, confidently presenting an inferred associative link as a strict causal mechanism could lead to misguided real-world decisions.

The authors should add a Broader Impact Statement that explicitly acknowledges the risks of relying on LLM-extracted "causality." This section should discuss the necessity of human-in-the-loop verification and suggest potential system guardrails, such as presenting inferred links with calibrated uncertainty estimates or hedging, before such a framework is deployed in critical environments.

**Claims And Evidence:**

No

**Claims Explanation:**

While the empirical results and ablations are promising, the current experimental design introduces significant confounds that obscure whether the performance gains stem from the SARG architecture or other factors.

Specifically, the evaluation restricts all methods to "gold documents." This artificial constraint neutralizes the primary advantage of global graph-based methods (which excel at discovering and aggregating scattered evidence across a vast corpus) and does not reflect a realistic end-to-end RAG deployment. Furthermore, the use of different backbone models across the evaluated methods (e.g., utilizing GPT-4o-mini for baseline graph methods while using smaller open-weight models for SARG in some reporting blocks) prevents a clean, apples-to-apples comparison.

Additionally, the foundational claim that SARG extracts accurate "causal" relationships is not directly supported by evidence. The paper lacks intermediate extraction-quality metrics (e.g., precision/recall of the extracted triples against human annotations, or entity resolution accuracy), leaving it unclear if the model is genuinely capturing causal logic or merely over-generating associative links that conveniently bridge context gaps.

**Requested Changes:**

***Critical to securing a recommendation for acceptance:***

- Standardized Baselines: Provide a "same-backbone" comparison to clearly isolate architectural gains from model capacity.

- End-to-End Evaluation: Include an evaluation setting that uses a standard retriever (dense or sparse) over a full corpus with distractors, rather than restricting the context solely to gold documents. Report retriever recall alongside downstream reasoning accuracy.

- Extraction Quality Metrics: Provide quantitative metrics for the intermediate graph construction. Specifically, report the precision and recall of the extracted causal/conditional triples against human annotations on a representative subset, and detail the accuracy/thresholds of the entity resolution process.

- Clarify Timing Discrepancies: Resolve the conflicting timing statements in the text (e.g., the 72–77 seconds "construction" time mentioned vs. the ~1.2 seconds per-query "inference construction"). Provide a clear, per-query wall-clock breakdown covering extraction, traversal, and generation, alongside average token costs.

***To strengthen the work:***

- Dataset and Code Release: Release the custom domain-specific datasets (Bitcoin Price, Gaucher Disease), the synthetic training data for the direction classifier, all prompts, and the code. Given the small sample sizes (20-25 documents), public release is highly recommended for reproducibility.

- Expanded Classifier Evaluation: The direction classifier reports 99% accuracy on a synthetic set. Provide a more detailed breakdown of its performance on the "Sim-to-Real" holdout set (e.g., confusion matrices by dataset/domain).

- Related Work: Expand the related work section to discuss recent query-time structuring and hybrid Graph-RAG methods to better position SARG's specific niche.

---

> ### Author Response · Authors · 2026-03-31
> **Response to Reviewer vf5P: Overview and RC1 (Standardized Baselines)**
>
> Thank you for your careful evaluation of the paper and recognizing the contributions of our work. We appreciate your assessment of the work's strengths: (1) **Architectural Novelty**: the post-retrieval, on-demand graph construction preserves RAG's flexibility while introducing the interpretability of graph-based reasoning; (2) **Design Choices**: enforcing a reasoning-specific schema controls graph noise relative to generic OpenIE, and the distilled direction classifier offers a principled systems optimization that controls branching without repeated LLM inference costs; and (3) **Interpretability**: by serializing explicit reasoning chains, the method produces traceable, step-by-step justifications grounded in retrieved evidence.
>
> We address the weaknesses and requested changes below, beginning with the requested changes.
>
> ---
>
> ## RC1. Standardized Baselines
>
> *Provide a "same-backbone" comparison to clearly isolate architectural gains from model capacity.*
>
> We agree that isolating architectural gains from model capacity is important. In our original submission, we evaluated graph-based baselines with GPT-4o-mini, while SARG was evaluated with open-source models. While we included a GPT-4o-mini reference block for SARG to enable direct comparison, we acknowledge that a fully standardized evaluation on a single backbone makes the differences cleaner.
>
> To address this, we re-evaluated all graph-based methods with Qwen2.5-7B-Instruct backbone. We report standard EM and F1 for HotPotQA and MuSiQue multi-hop QA benchmarks and use the G-Eval framework, implemented via the DeepEval library, to evaluate the quality of responses for custom Bitcoin and Gaucher datasets. This framework evaluates the output via 3 metrics: (1) **Faithfulness**: measuring whether the generated reasoning steps and answers are factually grounded in the retrieved context; (2) **Synthesis Accuracy**: assessing whether the final conclusion semantically aligns with the gold answer; and (3) **Reasoning Logic**: which checks that the reasoning trajectory is sound, and that the final conclusion is derived from the established premises.
>
> | Method | HPQA EM | HPQA F1 | MS EM | MS F1 | BTC-F | BTC-S | BTC-R | GAU-F | GAU-S | GAU-R |
> |---|---|---|---|---|---|---|---|---|---|---|
> | GraphRAG | 22.00 | 33.59 | 4.10 | 11.83 | 92.31 | 39.03 | 79.81 | 95.67 | 58.29 | 77.66 |
> | HippoRAG | 31.60 | 49.21 | 20.20 | 29.27 | 87.00 | 46.57 | 75.41 | 92.00 | 60.39 | 80.52 |
> | StructRAG | 27.20 | 38.61 | 17.20 | 24.78 | **95.72** | 55.93 | 74.48 | 98.52 | 67.87 | 80.22 |
> | HyperGraphRAG | 7.60 | 16.33 | 3.00 | 8.67 | 95.28 | 52.62 | 79.86 | 97.64 | 64.73 | 84.93 |
> | **SARG** | **51.24** | **64.13** | **27.56** | **40.83** | 94.83 | **65.00** | **86.72** | **99.71** | **73.58** | **91.60** |
>
> *Table 1. Performance comparison between SARG and Graph-based RAG methods using Qwen2.5-7B-Instruct.*
>
> SARG outperforms all baselines on every benchmark and metric with the exception of BTC-Faithfulness, where StructRAG and HyperGraphRAG score marginally higher. We attribute these to their conservative generation style on this dataset, as both methods produce shorter responses that are less likely to introduce unfaithful claims, but at the cost of substantially lower synthesis and reasoning logic scores.
>
> Our results are consistent with the original findings: SARG's advantage over global graph-based methods holds regardless of backbone model, confirming that the performance gains stem from architectural design.

---

> ### Author Response · Authors · 2026-03-31
> **Response to Reviewer vf5P: RC2 (End-to-End Evaluation) & RC3 (Extraction Quality)**
>
> ## RC2. End-to-End Evaluation
>
> *Include an evaluation setting that uses a standard retriever (dense or sparse) over a full corpus with distractors, rather than restricting the context solely to gold documents. Report retriever recall alongside downstream reasoning accuracy.*
>
> Thank you for your suggestion. Our original evaluation adopted an oracle setting, where we assumed that the gold supporting documents were already retrieved in order to isolate the contribution of the reasoning layer from retrieval noise. We agree that in realistic deployments, the retrieved context may be incomplete or contain distractors, and it is important to verify that SARG's reasoning gains are not an artifact of this setting.
>
> To address this, we conducted end-to-end open-retrieval experiments using bge-large-en-v1.5 as the retriever (top-5) with Qwen2.5-7B-Instruct as the generator across 500q per dataset (MuSiQue, HotPotQA). Retrieval recall@5 is 0.9450 on HotPotQA and 0.7447 on MuSiQue, confirming that the evaluation operates under realistic retrieval noise. We compare methods that operate directly over retrieved passages (Direct, RAG, CoT, SARG); global graph baselines are excluded as they derive their strengths from pre-indexing the corpus offline.
>
> | Method | HotPotQA EM | HotPotQA F1 | MuSiQue EM | MuSiQue F1 |
> |---|---|---|---|---|
> | Direct | 1.40 | 16.02 | 1.20 | 5.10 |
> | RAG | 35.20 | 42.28 | 13.80 | 21.50 |
> | CoT | 31.00 | 47.04 | 20.40 | 26.98 |
> | **SARG** | **47.80** | **60.92** | **23.40** | **31.44** |
>
> SARG's advantage is larger on HotPotQA, where retrieval recall is higher, suggesting that the method extracts maximum value from correctly retrieved evidence. Even on MuSiQue under substantial retrieval noise, SARG maintains a clear lead, confirming that the contribution is architectural — structured reasoning over whatever context is available — rather than dependence on the oracle setting.
>
> ---
>
> ## RC3. Extraction Quality Metrics
>
> *Provide quantitative metrics for the intermediate graph construction. Specifically, report the precision and recall of the extracted causal/conditional triples against human annotations on a representative subset, and detail the accuracy/thresholds of the entity resolution process.*
>
> We appreciate the reviewer raising this point. To address this, we conducted two complementary annotation studies on a representative subset of 83 triples drawn from the complete extraction output of 4 randomly sampled documents (2 Bitcoin Price, 2 Gaucher Disease).
>
> **Study 1: Factual correctness and causal validity.** Two independent annotators labeled each SARG-extracted triple on two dimensions: (1) *is_correct*: whether the source text supports the triple factually, and (2) *is_causal*: whether the relation is genuinely inferential rather than merely associative. Both annotators marked 100% of triples as factually correct. 88.6% were independently judged as genuinely causal or conditional rather than associative (Cohen's κ = 0.684, indicating substantial agreement). This directly addresses the concern that SARG may be over-generating associative links.
>
> **Study 2: Semantic Precision and Recall.** To provide quantitative coverage metrics, we evaluate precision, recall, and F1 using LLM-as-a-Judge (GPT-4o-mini) with bipartite Hungarian matching, following the soft-matching methodology advocated by Jiang et al. (2024) for open generative relation extraction. Against the union of both annotators' extractions as the gold set (union includes all triples identified by either annotator regardless of agreement), SARG achieves **P=0.816, R=0.681, F1=0.743**. We note these figures represent a conservative lower bound, as the human gold set includes some attributive facts that SARG's schema intentionally excludes.
>
> Taken together, these results confirm that SARG's extraction pipeline produces factually grounded, genuinely inferential triples, and that the reasoning-specific schema performs meaningful semantic filtering rather than over-extraction.
>
> > [1] Pengcheng Jiang, Jiacheng Lin, Zifeng Wang, Jimeng Sun, and Jiawei Han. "GenRES: Rethinking Evaluation for Generative Relation Extraction in the Era of Large Language Models." NAACL 2024. https://aclanthology.org/2024.naacl-long.155.pdf

---

> ### Author Response · Authors · 2026-03-31
> **Response to Reviewer vf5P: RC4 (Timing) & RC5 (Code Release) & RC6 (Classifier Evaluation) & RC7 (Related Work)**
>
> ## RC4. Clarify Timing Discrepancies
>
> *Resolve the conflicting timing statements in the text (e.g., the 72–77 seconds "construction" time mentioned vs. the ~1.2 seconds per-query "inference construction"). Provide a clear, per-query wall-clock breakdown covering extraction, traversal, and generation, alongside average token costs.*
>
> We appreciate the opportunity to clarify. At a high level, SARG's construction and inference construction measure two different phases of the pipeline.
>
> **Timing.** The 72-77s refers to **one-time graph construction cost**: extracting triples from retrieved documents, and assembling the KG from the document set. This is a one-time offline step per document corpus, analogous to the indexing phase in global methods (which take 361–557 seconds for the same task). ~1.2 seconds refers to the **per-query inference construction cost**: node matching, direction classification, and traversal over the already-constructed graph at query time. The total per-query wall-clock time is therefore ~1.2s (traversal) + ~7-10s (generation), which can be found in Figure 4.
>
> **Token Cost Clarification.** We provide a full cost breakdown to confirm that structured reasoning introduces negligible overhead. Graph extraction is the only LLM-intensive step, requiring ~65K input / 12K output tokens for Bitcoin and ~110K input / 16K output tokens for Gaucher, incurred once per corpus. At query time, steps 1-3 of the pipeline (KeyBERT concept extraction, DistilBERT direction classification, beam search traversal) are local, with zero API cost. Only the final answer generation step calls an LLM, consuming ~1200 input / ~80 output tokens/query. At GPT-4o-mini rates (\$0.15/1M input, \$0.60/1M output), the total end-to-end cost is \$0.022 for Bitcoin and \$0.032 for Gaucher, confirming that the reasoning layer adds minimal cost relative to standard RAG generation calls. We will improve the writing in the revised paper to make it clear.
>
> ---
>
> ## RC5. Dataset and Code Release
>
> *Release the custom domain-specific datasets (Bitcoin Price, Gaucher Disease), the synthetic training data for the direction classifier, all prompts, and the code. Given the small sample sizes (20-25 documents), public release is highly recommended for reproducibility.*
>
> All artifacts, such as the datasets, direction classifier model and training data, extraction prompts, and full pipeline code, have already been prepared for release. We commit to releasing all materials upon acceptance and will add the link to the GitHub repo in the camera-ready version.
>
> ---
>
> ## RC6. Expanded Classifier Evaluation
>
> *The direction classifier reports 99% accuracy on a synthetic set. Provide a more detailed breakdown of its performance on the "Sim-to-Real" holdout set (e.g., confusion matrices by dataset/domain).*
>
> We provide per-dataset confusion matrices on the Sim-to-Real holdout (n=50 per dataset, rows = gold GPT-4o-mini labels, columns = DistilBERT predictions):
>
> **HotPotQA (96% agreement, 2 errors):**
>
> | | Fwd | Bwd | Bidir |
> |---|---|---|---|
> | **Fwd** | 17 | 0 | 1 |
> | **Bwd** | 0 | 17 | 0 |
> | **Bidir** | 0 | 1 | 14 |
>
> **MuSiQue (88% agreement, 6 errors):**
>
> | | Fwd | Bwd | Bidir |
> |---|---|---|---|
> | **Fwd** | 15 | 1 | 1 |
> | **Bwd** | 0 | 17 | 1 |
> | **Bidir** | 1 | 2 | 12 |
>
> The results confirm strong generalization across both datasets. Backward queries are classified with near-perfect recall (17/17 on both datasets), consistent with their lexically distinctive cause-seeking language. Errors are mostly concentrated in the Bidirectional class, where relational queries can superficially resemble forward consequence-tracing — for example, "What other district is found in the same county as Gmina Stezyca?" shares surface structure with forward queries but expresses a symmetric relationship. This is the expected failure mode and does not affect the dominant Forward/Backward classes that drive SARG's traversal.
>
> ---
>
> ## RC7. Related Work
>
> *Expand the related work section to discuss recent query-time structuring and hybrid Graph-RAG methods to better position SARG's specific niche.*
>
> We will substantially expand the related work section in the revision to position SARG along two axes: when the graph is built (offline corpus-level vs. online query-time) and what it encodes (generic SPO facts vs. reasoning-specific relations), covering recent methods such as RAS (Jiang et al., 2025) and Structure-R1 (Wu et al., 2025).
>
> > [2] Pengcheng Jiang, Lang Cao, Ruike Zhu, Minhao Jiang, Yunyi Zhang, Jimeng Sun, and Jiawei Han. "RAS: Retrieval-And-Structuring for Knowledge-Intensive LLM Generation." 2025. https://arxiv.org/abs/2502.10996
>
> > [3] Junlin Wu, Xianrui Zhong, Jiashuo Sun, Bolian Li, Bowen Jin, Jiawei Han, and Qingkai Zeng. "Structure-R1: Dynamically Leveraging Structural Knowledge in LLM Reasoning through Reinforcement Learning." 2025. https://arxiv.org/abs/2510.15191

---

> ### Author Response · Authors · 2026-03-31
> **Response to Reviewer vf5P: Weaknesses W1-W3**
>
> **W1. Experimental Confounds:** *Comparing methods using different backbone models and artificially restricting the context to "gold documents" undermines the fairness of the comparisons against global graph baselines.*
>
> As shown in our response to RC2, we have conducted new end-to-end open-retrieval experiments that evaluate SARG under realistic retrieval noise, and in RC1, we provide a fully standardized backbone comparison across all methods. Together, these results confirm that SARG's reasoning gains are neither due to the oracle document setting nor a consequence of model capacity differences, but they reflect genuine architectural advantages in post-retrieval structured reasoning.
>
> **W2. Missing Intermediate Evaluations:** *The paper lacks quantitative evaluation of the underlying LLM-based triple extraction fidelity and entity resolution accuracy.*
>
> As detailed in our response to RC3, we conducted two complementary studies to verify extraction fidelity: (1) a human annotation study confirming 100% factual correctness and 88.6% causal validity (κ = 0.684), and (2) a quantitative LLM-as-a-Judge evaluation yielding P = 0.816, R = 0.681, F1 = 0.743 against human reference extractions.
>
> **W3. Evaluation Robustness:** *Heavy reliance on LLM-as-a-judge for very small custom datasets introduces risks of bias and overfitting.*
>
> We share the reviewer's concern, which is why our assessment is deliberately multi-faceted. The domain-specific G-Eval scores are corroborated by human evaluations, where SARG is preferred 2.7× more than the strongest baselines. We also include bootstrap standard deviations in the revised version of the manuscript.

---

### Review · Reviewer_4kfP · 2026-03-03

**Summary Of Contributions:**

The authors introduce a post-retrieval framework designed to improve multi-hop reasoning over retrieved passages in RAG pipelines by making cross-document structure explicit. The core method constructs a knowledge graph on-the-fly at inference time from retrieved documents by first extracting reasoning-specific triples via few-shot prompting, mapping query concepts to graph nodes and using a DistilBERT classifier to determine edge direction, then performing beam search to discover chains and serializing them into a structured prompt. Evaluation on HotpotQA, MuSiQue, and domain-specific datasets show improvement over various “flat-context” and graph-based RAG baselines.

Strengths:
1. The framework presented is well-structured and methodically described, with solid experimentation.
2. The ablations are motivated well and match the claims, testing components of the process and the results are directionally consistent with the method’s intent — the intervention introduced is aligned well to the failure mode it targets, namely cross-document synthesis.
3. The inclusion of human evaluation in addition to automated evaluation is welcome, and the qualitative case study provides some valuable intuition regarding the method’s behavior regarding the construction of multi-hop links.

Weaknesses:
1. Evaluation is only performed using gold documents rather than documents from an actual retrieval. While this is fine to isolate the reasoning question, it makes it unclear how SARG performs under imperfect / realistic retrieval noise.
2. The claim that instruction-tuned LLMs can “reliably extract high-precision … triples” is motivated but not backed by a direct evaluation of extraction (perhaps with precision/recall against human labels)
3. While the domain-specific datasets are appreciated, they are very small (20 and 25 samples, respectively, as per Table 2), which makes the G-Eval scores highly sensitive to each question’s outcome and the human evaluation is similarly limited (N=30).

**Audience:**

Yes

**Audience Explanation:**

The problem tackled in this work — improving multi-hop reasoning in RAG systems — is highly relevant to the NLP and language modeling research community as well as to practitioners deploying LLM-based systems. The retrieval-agnostic mechanism deisgned in this work is appealing for real-world settings as a plug-in with existing retrieval setups. Local structuring after retrieval is a practical alternative to corpus-wide indexing, offering a compelling approach of interest. Explicit chain serialization is valuable for interpretability, enhancing the possibility of adoption by practitioners.

**Broader Impact Concerns:**

Given that the paper emphasizes interpretable and traceable reasoning chains, and aims to induce impact in high-stakes settings like medicine and finance, a brief discussion regarding the risk of overreliance on the method’s ouputs would be appreciated — especially given the extraction step could introduce intermediate errors, and the clearly lower faithfulness score on the Bitcoin Price dataset.

**Claims And Evidence:**

Yes

**Claims Explanation:**

While most core claims appear to be largely substantiated, others require further evidence to form a complete and convincing narrative. Please see the weaknesses above and the requested changes.

**Requested Changes:**

1. The baselines for graph-based methods use gpt-4o-mini, rather than any of the smaller open-weight models. While it is understood that this is “consistent with their original implementations”, it is important to have a standardized comparison across all methods to confirm that the advantage of SARG comes from the architecture, rather than model-specific behavior, when it comes to outperforming those baselines.
2. Evaluation with a sparse retriever like BM25 and/or a dense retriever (e.g. DPR) while varying top-k would be useful to ensure the method’s robustness and study its behavior under retrieval noise.
3. Without confidence intervals or standard deviations, it’s difficult to draw broader conclusions about those findings with the domain-specific datasets. Qualitative error analysis would be also useful to reinforce the aggregate scores. Addressing these will substantiate the inclusion of those findings.
4. Understanding or categorizing failure modes when faithfulness decreased on Bitcoin Price, especially with such a small set, would be valuable to inform practitioners — that is, whether this is a result of extraction errors, or from graph construction, or chain serialization, or another cause.
5. Specifying all hyperparameters (e.g. the node matching threshold $\tau$, number of few-shot exemplars, as well as chunking strategy and chunk size.

---

> ### Author Response · Authors · 2026-03-31
> **Response to Reviewer 4kfP: Overview, RC1 (Standardized Baselines)**
>
> We thank the reviewer for their thorough evaluation of our paper and for recognizing our work's contributions. We appreciate your assessment of the paper's strengths: (1) the framework is well-structured and methodically described, with solid experimentation; (2) the ablations are motivated well and match the claims, testing components of the process results; and (3) the inclusion of human evaluation in addition to automated evaluation, and the qualitative case study provides some valuable intuition regarding the method's behavior regarding the construction of multi-hop links.
>
> We are also grateful for the constructive feedback on the paper's limitations. We address each requested change and weakness below.
>
> ---
>
> ## RC1. Standardized Baselines
>
> *The baselines for graph-based methods use gpt-4o-mini, rather than any of the smaller open-weight models. While it is understood that this is "consistent with their original implementations", it is important to have a standardized comparison across all methods to confirm that the advantage of SARG comes from the architecture, rather than model-specific behavior, when it comes to outperforming those baselines.*
>
> We agree that isolating architectural gains from model capacity is important. In our original submission, we evaluated graph-based baselines with GPT-4o-mini, while SARG was evaluated with open-source models. While we included a GPT-4o-mini reference block for SARG to enable direct comparison, we acknowledge that a fully standardized evaluation on a single backbone makes the differences cleaner.
>
> To address this, we re-evaluated all graph-based methods with Qwen2.5-7B-Instruct backbone. We report standard EM and F1 for HotPotQA and MuSiQue multi-hop QA benchmarks and use the G-Eval framework, implemented via the DeepEval library, to evaluate the quality of responses for custom Bitcoin and Gaucher datasets. This framework evaluates the output via 3 metrics: (1) **Faithfulness**: measuring whether the generated reasoning steps and answers are factually grounded in the retrieved context; (2) **Synthesis Accuracy**: assessing whether the final conclusion semantically aligns with the gold answer; and (3) **Reasoning Logic**: which checks that the reasoning trajectory is sound, and that the final conclusion is derived from the established premises.
>
> | Method | HPQA EM | HPQA F1 | MS EM | MS F1 | BTC-F | BTC-S | BTC-R | GAU-F | GAU-S | GAU-R |
> |---|---|---|---|---|---|---|---|---|---|---|
> | GraphRAG | 22.00 | 33.59 | 4.10 | 11.83 | 92.31 | 39.03 | 79.81 | 95.67 | 58.29 | 77.66 |
> | HippoRAG | 31.60 | 49.21 | 20.20 | 29.27 | 87.00 | 46.57 | 75.41 | 92.00 | 60.39 | 80.52 |
> | StructRAG | 27.20 | 38.61 | 17.20 | 24.78 | **95.72** | 55.93 | 74.48 | 98.52 | 67.87 | 80.22 |
> | HyperGraphRAG | 7.60 | 16.33 | 3.00 | 8.67 | 95.28 | 52.62 | 79.86 | 97.64 | 64.73 | 84.93 |
> | **SARG** | **51.24** | **64.13** | **27.56** | **40.83** | 94.83 | **65.00** | **86.72** | **99.71** | **73.58** | **91.60** |
>
> *Table 1. Performance comparison between SARG and Graph-based RAG methods using Qwen2.5-7B-Instruct.*
>
> SARG outperforms all baselines on every benchmark and metric with the exception of BTC-Faithfulness, where StructRAG and HyperGraphRAG score marginally higher. We attribute the marginally higher faithfulness scores of StructRAG and HyperGraphRAG on Bitcoin Price to two complementary factors. First, both methods produce shorter, more conservative responses that are less likely to introduce unfaithful claims at the cost of substantially lower synthesis and reasoning logic scores. Second, as our error analysis reveals (see RC4), SARG's chain scoring function currently ranks thematically relevant chains over more precise ones containing specific factual evidence (e.g., dollar amounts, named actors). The generator then produces responses that the faithfulness evaluator penalizes because specific claims cannot be grounded in the selected chain context. We believe incorporating specificity-aware reranking into the chain scoring function would close this narrow gap while preserving SARG's stronger synthesis and reasoning performance.
>
> Our results are consistent with the original findings: SARG's advantage over global graph-based methods holds regardless of backbone model, confirming that the performance gains stem from architectural design.

---

> ### Author Response · Authors · 2026-03-31
> **Response to Reviewer 4kfP: RC2 (Retrieval Noise), RC3 (Confidence Intervals), RC4 (Failure Analysis) & RC5 (Hyperparameters)**
>
> ## RC2. Evaluation Under Retrieval Noise
>
> *Evaluation with a sparse retriever like BM25 and/or a dense retriever (e.g. DPR) while varying top-k would be useful to ensure the method's robustness and study its behavior under retrieval noise.*
>
> Thank you for your suggestion. Our original evaluation adopted an oracle setting, where we assumed that the gold supporting documents were already retrieved in order to isolate the contribution of the reasoning layer from retrieval noise. We agree that in realistic deployments, the retrieved context may be incomplete or contain distractors, and it is important to verify that SARG's reasoning gains are not an artifact of this setting.
>
> To address this, we conducted end-to-end open-retrieval experiments using bge-large-en-v1.5 as the retriever (top-5) with Qwen2.5-7B-Instruct as the generator across 500q per dataset (MuSiQue, HotPotQA). Retrieval recall@5 is 0.9450 on HotPotQA and 0.7447 on MuSiQue, confirming that the evaluation operates under realistic retrieval noise. We compare methods that operate directly over retrieved passages (Direct, RAG, CoT, SARG); global graph baselines are excluded as they derive their strengths from pre-indexing the corpus offline.
>
> | Method | HotPotQA EM | HotPotQA F1 | MuSiQue EM | MuSiQue F1 |
> |---|---|---|---|---|
> | Direct | 1.40 | 16.02 | 1.20 | 5.10 |
> | RAG | 35.20 | 42.28 | 13.80 | 21.50 |
> | CoT | 31.00 | 47.04 | 20.40 | 26.98 |
> | **SARG** | **47.80** | **60.92** | **23.40** | **31.44** |
>
> SARG's advantage is larger on HotPotQA, where retrieval recall is higher, suggesting that the method extracts maximum value from correctly retrieved evidence. Even on MuSiQue under substantial retrieval noise, SARG maintains a clear lead, confirming that the contribution is architectural — structured reasoning over whatever context is available — rather than dependence on the oracle setting.
>
> ---
>
> ## RC3. Confidence Intervals and Error Analysis
>
> *Without confidence intervals or standard deviations, it's difficult to draw broader conclusions about those findings with the domain-specific datasets. Qualitative error analysis would be also useful to reinforce the aggregate scores. Addressing these will substantiate the inclusion of those findings.*
>
> We thank the reviewer for their comment, we agree that statistical analysis is important for drawing reliable conclusions from our small datasets. We have added standard deviations to all domain-specific results in the revised Table 1. The GPT-4o-mini backbone is shown below.
>
> | Method | BP-F | BP-S | BP-L | GD-F | GD-S | GD-L |
> |---|---|---|---|---|---|---|
> | GraphRAG | 91.91 (0.05) | 52.73 (0.15) | 82.33 (0.06) | 97.46 (0.03) | 74.31 (0.16) | 92.78 (0.02) |
> | HippoRAG2 | 91.54 (0.12) | 64.29 (0.08) | 84.70 (0.07) | 98.49 (0.05) | 72.52 (0.10) | 93.55 (0.01) |
> | HyperGraphRAG | 91.83 (0.04) | 53.65 (0.09) | 84.36 (0.04) | 97.51 (0.02) | 70.00 (0.05) | 93.34 (0.03) |
> | StructRAG | **92.26 (0.08)** | 45.00 (0.02) | 86.12 (0.04) | 98.22 (0.04) | 70.09 (0.06) | 91.00 (0.01) |
> | **SARG** | 91.90 (0.07) | **70.40 (0.04)** | **87.90 (0.05)** | **99.70 (0.01)** | **83.50 (0.07)** | **96.10 (0.02)** |
>
> ---
>
> ## RC4. Faithfulness Failure Analysis
>
> *Understanding or categorizing failure modes when faithfulness decreased on Bitcoin Price, especially with such a small set, would be valuable to inform practitioners — that is, whether this is a result of extraction errors, or from graph construction, or chain serialization, or another cause.*
>
> We thank the reviewer for this suggestion. We conducted a systematic error analysis on the Bitcoin Price dataset, examining all cases where faithfulness dropped. All three errors share the same root cause: the chain scoring function ranks thematically relevant chains over more precise ones containing the specific factual evidence (e.g. dollar amounts, named actors, specific events) that the gold answer requires. The generator then either stays vague or produces a generic summary that the faithfulness evaluator penalizes because the specific claims cannot be grounded in the selected chain context.
>
> This points to a targeted improvement: incorporating specificity-aware reranking into the chain scoring function, so that chains containing fine-grained factual evidence are preferred over thematically similar but coarser alternatives. We will include this analysis in the revised manuscript.
>
> ---
>
> ## RC5. Hyperparameter Specification
>
> *Specifying all hyperparameters (e.g. the node matching threshold, number of few-shot exemplars, as well as chunking strategy and chunk size).*
>
> The hyperparameter values are as follows: node matching threshold τ = 0.75, beam width = 3, top-k chains = 3, max starting nodes = 3, max depth = 3 hops (BP/GD). For baseline RAG: chunk size = 400 words, overlap = 50 words, retriever = all-MiniLM-L6-v2, top-k chunks = 5. We will add a complete hyperparameter table to the appendix in the revision.

---

> ### Author Response · Authors · 2026-03-31
> **Response to Reviewer 4kfP: Weaknesses W1–W3**
>
> ## Addressing Weaknesses
>
> **W1. Evaluation under realistic retrieval noise:** This concern is directly addressed by the new open-retrieval experiments presented in RC2, where SARG's reasoning gains hold under realistic retrieval noise with bge-large-en-v1.5 at top-5 recall rates of 0.9450 (HotPotQA) and 0.7447 (MuSiQue).
>
> **W2. Direct evaluation of extraction quality:** We conducted two complementary annotation studies on 83 triples: (1) a human annotation study confirming 100% factual correctness and 88.6% causal validity (Cohen's κ = 0.684), and (2) a quantitative LLM-as-a-Judge evaluation yielding P = 0.816, R = 0.681, F1 = 0.743 against human reference extractions, following the soft-matching methodology of Jiang et al. [1] for open generative relation extraction.
>
> > [1] Pengcheng Jiang, Jiacheng Lin, Zifeng Wang, Jimeng Sun, and Jiawei Han. "GenRES: Rethinking Evaluation for Generative Relation Extraction in the Era of Large Language Models." NAACL 2024. https://aclanthology.org/2024.naacl-long.155.pdf
>
> **W3. Small domain-specific datasets:** We share this concern, which is why we have added bootstrap standard deviations to all G-Eval scores (see RC3) and conducted a qualitative error analysis categorizing failure modes on Bitcoin Price (see RC4). We emphasize that the domain-specific datasets are deliberately small — they are deeply annotated case studies requiring domain expertise. The large-scale evidence comes from HotPotQA and MuSiQue (500 questions each) with EM/F1 evaluation. The domain-specific datasets complement these by demonstrating generalization to longer-form explanatory reasoning tasks where short-answer metrics are inadequate, and where SARG is preferred 2.7× over the strongest baseline in human evaluation.

---

### Review · Reviewer_LaSv · 2026-04-17

**Summary Of Contributions:**

The paper proposes SARG, a post-processing step for RAG that aims to help LLMs reason better over retrieved documents. Instead of directly providing the LLM with a flat set of retrieved chunks and requiring it to connect the relevant information implicitly, SARG first builds a knowledge graph over the retrieved documents before generation. The experimental results suggest that this structured post-processing can improve reasoning quality over flat-context RAG baselines as well as other graph-enhanced baselines.

**Strength:**

The motivation of the method is clear, a post processing step on the retrieved sources should enable LLM to better reason about the sources.

The paper’s experiments are informative and promising, and the new datasets are interesting.

**Weakness:**

The new custom datasets are rather small (20 and 25 test instances), so the results aren’t as conclusive.

The presentation of the paper can be improved. In particular, the main stages of SARG are not very clear: the abstract describes 3 stages, while the main text describes 4. Furthermore, in Section 3 it is not immediately clear which subsection corresponds to which stage.

Some claims are not fully supported, see below.

**Audience:**

Yes

**Audience Explanation:**

The question of how to improve reasoning in RAG systems after retrieval, without changing the retriever itself, is of clear interest.

**Claims And Evidence:**

No

**Claims Explanation:**

The paper provides promising evidence that SARG improves over flat-context RAG baselines on the reported tasks, and the experimental section is generally informative.

However, as it currently stands, I do not think all of the paper’s claims are fully supported.

First, one of the main research questions is whether SARG performs better than expert annotation. However, the evidence for this claim appears to be limited to a single case study, which is not sufficient to draw a strong conclusion.

Second, compared with flat-context RAG baselines, SARG introduces substantial additional inference-time processing. It remains unclear how much of the gain over flat baselines comes from the proposed structured reasoning mechanism itself, as opposed to the additional post-retrieval computation. I would have liked to see stronger controlled comparisons or discussion on this point.

**Requested Changes:**

Please clarify the main components of the pipeline and present them more consistently.

Can the paper say more about the additional computational overhead, and more clearly separate the benefit of the structured reasoning mechanism from the benefit of extra inference-time computation?

Minor:
The abbreviation OpenIE is used before it is introduced in Section 2.

---

> ### Author Response · Authors · 2026-04-19
> **Response to Reviewer LaSv: Weaknesses W1-W2**
>
> We thank reviewer LaSv for careful evaluation and for recognizing the contributions of our work. We appreciate the assessment of our strengths: (1) Clear Motivation: that a post-retrieval structuring step over retrieved sources should enable LLMs to reason more effectively, which is precisely the gap SARG is designed to fill; (2) Informative Experiments: that the experimental design is promising and provides useful evidence for SARG's effectiveness; and (3) Novel Datasets: that the custom domain-specific benchmarks contribute interesting evaluation scenarios for structured reasoning in specialized domains. We address the weaknesses and requested changes below.
>
> ## W1. Small Custom Datasets
>
> We acknowledge that the custom dataset sizes are limited. However, we want to emphasize that our primary empirical evidence rests on two established multi-hop benchmarks (HotPotQA and MuSiQue), where SARG achieves the highest global EM and F1. The custom datasets serve a complementary role in our eval suite, as they evaluate long-form explanatory reasoning in specialized domains, requiring synthesis of technical information across documents rather than short factual answers.
>
> To ensure that conclusions drawn from these smaller sets are robust, our evaluation is deliberately multi-faceted. The automatic G-Eval scores are independently corroborated by a blinded human evaluation study (Section 5.3, Figure 2), in which three domain-expert annotators rated SARG highest across both datasets, achieving win rates of 32.2% (BP) and 35.6% (GD), 2.7x higher than the strongest baseline, with substantial inter-annotator agreement (Krippendorff's α = 0.71). In the revised manuscript, we will explicitly frame these datasets as targeted domain-specific case studies rather than large-scale benchmarks.
> Additionally, to demonstrate that SARG's gains hold at scale under realistic conditions, we conducted end-to-end open-retrieval experiments using BGE-large-en-v1.5 as the retriever (top-5) with Qwen2.5-7B-Instruct as the generator across 500 questions per dataset. Retrieval recall@5 is 0.9450 on HotPotQA and 0.7447 on MuSiQue, confirming that the evaluation operates under realistic retrieval noise rather than gold-document conditions. We compare methods that operate directly over retrieved passages; global graph baselines are excluded as they derive their performance from pre-indexing the corpus offline.
>
> | Method | HPQA EM | HPQA F1 | MS EM | MS F1 |
> |--------|---------|---------|-------|-------|
> | Direct | 1.40 | 16.02 | 1.20 | 5.10 |
> | RAG | 35.20 | 42.28 | 13.80 | 21.50 |
> | CoT-RAG | 31.00 | 47.04 | 20.40 | 26.98 |
> | SARG | **47.80** | **60.92** | **23.40** | **31.44** |
>
> SARG outperforms all baselines under open retrieval, achieving +12.60 EM / +13.88 F1 over the next best method on HotPotQA and +3.00 EM / +4.46 F1 on MuSiQue. Notably, the performance trends are consistent with our gold-document results (Table 1), confirming that SARG's structured reasoning generalizes beyond controlled settings and that the architectural gains are not artifacts of small dataset size. We will include these results in the revised manuscript.
>
> ## W2. Presentation Inconsistency
>
> We thank the reviewer for catching this. We will revise the abstract to consistently reference four stages and add explicit stage labels to Section headers so that the mapping is immediately clear.

---

> ### Author Response · Authors · 2026-04-19
> **Response to Reviewer LaSv: RC1 (Expert Annotation) & RC2 (Computational Overhead) & RC3 (OpenIE Abbreviation)**
>
> ## RC1. Expert Annotation Claim
>
> We would like to clarify the intended scope of this claim. Section 5.4 is not designed to argue that SARG generally outperforms expert annotation. Rather, the case study validates that SARG's fully unsupervised extraction pipeline — requiring no domain-specific fine-tuning, manual annotation, or reinforcement learning — produces a knowledge graph of sufficient quality to support downstream multi-hop reasoning.
> The case study illustrates a specific failure mode of static expert annotation: the human-annotated KG from Antonucci et al. (2023) captures entity-level associations (e.g., ⟨susceptibility genes, FT1D⟩) but omits the procedural intermediate steps (FT1D → necessitates diagnosis → enables timely treatment) that are required to answer the query. SARG's reasoning-specific schema recovers these links because it explicitly targets causal and procedural dependencies rather than associative relations.
> To validate extraction quality more rigorously, we conducted two complementary annotation studies on a representative subset of 83 triples drawn from the complete extraction output of 4 randomly sampled documents (2 Bitcoin Price, 2 Gaucher Disease).
>
> Study 1: Factual correctness and causal validity: Two independent annotators labeled each SARG-extracted triple on two dimensions: (1) is_correct: whether the source text supports the triple factually, and (2) is_causal: whether the relation is genuinely inferential rather than merely associative. Both annotators marked 100% of triples as factually correct. 88.6% were independently judged as genuinely causal or conditional rather than associative (Cohen's κ = 0.684, indicating substantial agreement). This directly addresses the concern that SARG may be over-generating associative links.
>
> Study 2: Semantic Precision and Recall: To provide quantitative coverage metrics, we evaluate precision, recall, and F1 using LLM-as-a-Judge (GPT-4o-mini) with bipartite Hungarian matching, following the soft-matching methodology advocated by Jiang et al. (2024) [1] for open generative relation extraction. Against the union of both annotators' extractions as the gold set (union includes all triples identified by either annotator regardless of agreement), SARG achieves P=0.816, R=0.681, F1=0.743. We note these figures represent a conservative lower bound, as the human gold set includes some attributive facts that SARG’s schema intentionally excludes.
>
> Taken together, these results confirm that SARG's extraction pipeline produces factually grounded, genuinely inferential triples, and that the reasoning-specific schema performs meaningful semantic filtering.
>
> [1] Pengcheng Jiang, Jiacheng Lin, Zifeng Wang, Jimeng Sun, and Jiawei Han. "GenRES: Rethinking Evaluation for Generative Relation Extraction in the Era of Large Language Models." NAACL 2024. https://aclanthology.org/2024.naacl-long.155.pdf
>
> We will revise the framing to make this intent clearer and avoid any implication of broad superiority over expert annotation.
>
> ## RC2. Computational Overhead
>
> We appreciate the reviewer for bringing up this discussion and believe the paper has already provided two controlled comparisons that isolate this factor.
> First, our baselines include CoT-RAG (Table 1), which gives the flat-context baseline additional inference-time reasoning budget via chain-of-thought prompting, without any structural post-processing. SARG outperforms CoT-RAG across all benchmarks and backbones, demonstrating that the gains come from the structured representation rather than extra computation alone.
> Second, the graph-based baselines (GraphRAG, HippoRAG2, HyperGraphRAG, StructRAG) all introduce comparable or greater post-retrieval computation (e.g. graph construction, community summarization, schema inference) and SARG outperforms all of them. Our analysis in Section 5.5 and Figure 4 quantifies this directly: GraphRAG's inference construction exceeds 8s per query compared to SARG's ~1.2s, yet SARG achieves higher scores across all metrics. Together, these comparisons confirm that SARG's gains stem from how it organizes and serializes structure, not merely from additional computation.
> We will add a more prominent discussion to emphasize this point in the revision.
>
> ## RC3. OpenIE Abbreviation
>
> We thank the reviewer for catching this and will fix this in the revised version.